# Nodal band-off-diagonal superconductivity in twisted graphene superlattices

Maine Christos[1], Subir Sachdev ®[1] & Mathias S. Scheurer ®[2,3] ✉

The superconducting state and mechanism are among the least understood phenomena in twisted graphene systems. Recent tunneling experiments indicate a transition between nodal and gapped pairing with electron filling, which is not naturally understood within current theory. We demonstrate that the coexistence of superconductivity and flavor polarization leads to pairing channels that are guaranteed by symmetry to be entirely band-off-diagonal, with a variety of consequences: most notably, the pairing invariant under all symmetries can have Bogoliubov Fermi surfaces in the superconducting state with protected nodal lines, or may be fully gapped, depending on parameters, and the band-off-diagonal chiral $p$-wave state exhibits transitions between gapped and nodal regions upon varying the doping. We demonstrate that band-off-diagonal pairing can be the leading state when only phonons are considered, and is also uniquely favored by fluctuations of a time-reversal-symmetric intervalley coherent order motivated by recent experiments. Consequently, band-off-diagonal superconductivity allows for the reconciliation of several key experimental observations in graphene moiré systems.

The fascinating physics[1,2] of correlated graphene moiré superlattices, such as twisted-bilayer (TBG) and twisted-trilayer graphene (TTG), has generated extensive efforts to uncover the mysteries of their phase diagrams. Much progress has been made towards understanding their normal-state physics, including the correlated insulating phases[3–18] and the reset behavior[19,20]; the latter, which is believed to be associated with the onset of flavor polarization, appears in the same density range and can coexist with superconductivity[13,19–34]. However, the form and symmetry of the superconducting order parameter and the pairing glue are still unknown, despite significant theoretical efforts[27–30,33,35–47].

Tunneling conductance measurements taken within the superconducting state reveal a V-shaped density of states (DOS)[48,49] which can become $U$-shaped at other electron concentrations[49]. Setting aside the possibility of thermal fluctuations as origin[50], this is most naturally interpreted as a transition from nodal to fully gapped superconductivity. For a consistent microscopic theoretical understanding, this provides the following challenges: (i) electron–phonon coupling—a widely discussed[33,35–40] pairing mechanism in TBG and TTG—will typically mediate an entirely attractive interaction in the Cooper

channel, with the leading pairing state that transforms trivially under all symmetries and is thus fully gapped[51,52]. (ii) Even when the low-energy interactions favor an irreducible presentation (IR), e.g., $E$ of $C_3$, with nodal basis functions ($p$- or $d$-wave), the generically fully gapped chiral configuration wins over the nodal nematic one within mean-field. (iii) Even if we assume that the nodal state is energetically favored, e.g., due to significant corrections beyond mean-field[27,53–55], one is still left to explain why there is a transition to another, fully gapped superconductor upon changing the filling.

In this work, we show that the combination of flavor polarization and the representations of the symmetries in the flat bands of TBG and TTG allow for pairing channels that are completely off-diagonal in the flat bands and that such band-off-diagonal states can naturally reconcile all three key challenges (i–iii). More specifically, we find two distinct band-off-diagonal states: one of them transforms under the trivial representation $A$ of the system's point group $C_6$ (or one of $A_{1,2}$ of $D_6$ if we set the displacement field to zero) but can nonetheless have symmetry-protected nodal lines, akin to Bogoliubov Fermi surfaces discussed in refs. 56,57, see Fig. 1a–c for an intuitive visual explanation.

[1]Department of Physics, Harvard University, Cambridge, MA 02138, USA. [2]Institute for Theoretical Physics, University of Innsbruck, Innsbruck A-6020, Austria. [3]Institute for Theoretical Physics III, University of Stuttgart, 70550 Stuttgart, Germany. ✉e-mail: mathias.scheurer@itp3.uni-stuttgart.de

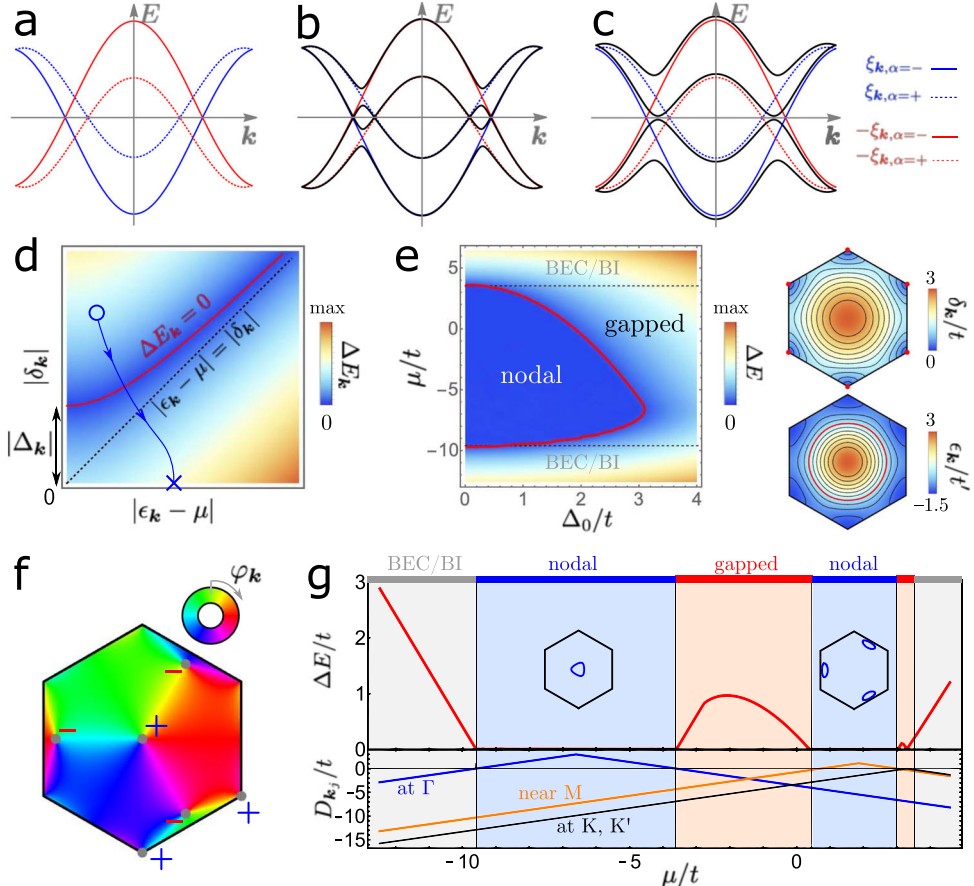

**Fig. 1 | Spectral properties of interband pairing.** While for band-diagonal pairing a small superconducting order parameter can immediately open up a gap as time-reversal symmetry guarantees that the associated avoided crossings [gray regions in a] in the Bogoliubov spectrum are at the Fermi level, this is not the case for band-off-diagonal pairing (**b**). Here, a sufficiently strong order-parameter value is required to establish a full gap, see **c**. Its $k$ dependence according to Eq. (3) is shown in **d**, where the red line indicates nodal points. If the band structure has Dirac points, there will be a point on the horizontal axis (blue cross). Consequently, if there is another momentum point located above the red line (blue circle),

continuity of the Hamiltonian implies a nodal point on any path connecting the two momenta. **e** Gap of the isotropic $A_2$ state and $\delta_k$, $\epsilon_k$ (zeros indicated in red) for the normal-state toy model defined in the text. BEC/BI refers to the Bose-Einstein condensate/band insulator limit. **f** Complex phase $\varphi_k = \arg(X_k + iY_k)$ for leading basis function with small subleading corrections. **g** Shows the gap of the chiral $p$-wave $E_2(1, i)$ state with $\Delta_0 = 1.5t$ and the value of $D_{k_j} := |\delta_{k_j}| - |\epsilon_{k_j} - \mu|$ for $k_j$ at the three symmetry-in-equivalent vortices in **f** as a function of $\mu$. We took $t' = -2.2t, t > 0$, in **b**, **d**.

The surprising possibility of the existence of such Bogliubov Fermi surfaces without an external magnetic field is unique to twisted graphene systems in that it follows as a direct consequence of both the symmetry and relative flatness of their normal-state bands. The second off-diagonal state transforms under a two-dimensional IR ($E_2$ of $C_6$). Its associated chiral state, $E_2(1, i)$, which is favored in the mean-field over the nematic one, has the unique property of exhibiting nodal lines or being fully gapped depending on the filling fraction, even when the order parameter is kept fixed. We supplement our general symmetry arguments and phenomenological models with Hartree-Fock (HF) calculations on the continuum model, studying a variety of different pairing mechanisms. We find that nodal band-off-diagonal pairing is favored by the optical $A_1$ and $B_1$ phonon modes and by fluctuations of a time-reversal symmetric intervalley coherent (T-IVC) state (the T-IVC state has Kekulé order on the graphene scale[58–60]). Evidence for the former has been provided by a recent photoemission study[61] while evidence for the latter comes from recent STM experiments[7]. Furthermore, also fluctuations of a time-reversal-symmetric sublattice polarized state (SLP+) are attractive in the band-off-diagonal channel (see Table 2 for a formal definition of the order parameters). We also show that fluctuations of both T-IVC and of a nematic, time-reversal symmetric IVC order[62] favor either the band-off-diagonal $A$ or an $E_1$ state with band-diagonal components, which may also be nodal; the

winner is determined by the relative amount of nematic IVC and T-IVC fluctuations.

## Results
### Possible pairing states
Let us begin by classifying the superconducting instabilities in graphene moiré systems in the limit where the low-energy bands are spin-polarized but allow for multiple bands. We denote the spinless low-energy fermionic creation operators by $c^\dagger_{k,\alpha,\eta}$ with momentum $k$ in valley $\eta = \pm$, and of band index $\alpha$ labeling the upper ($\alpha = +$) and lower ($\alpha = -$) quasi-flat bands. As a result of two-fold rotational symmetry, $C_{2z}$, along the out-of-plane ($z$) direction or effective spinless time-reversal symmetry, $\Theta$, the non-interacting band structure $\xi_{k,\alpha,\eta}$ obeys $\xi_{k,\alpha,\eta} = \xi_{-k,\alpha,-\eta} \equiv \xi_{\eta \cdot k,\alpha}$ and intervalley pairing is expected to dominate. A general pairing order parameter in the inter-valley channel couples as

$$\mathcal{H}_p = \sum_{k,\eta=\pm,\alpha,\alpha'} c^\dagger_{k,\alpha,\eta} \left(\Delta_{k,\eta}\right)_{\alpha,\alpha'} c^\dagger_{-k,\alpha',-\eta} + \text{H.c.,} \qquad (1)$$

where the order parameter $\Delta_{k,\eta} = -\Delta^T_{-k,-\eta}$ is a matrix in band space. The physical spin texture of the superconductor is entirely determined by the form of the underlying normal-state's polarization: if the spins are aligned in the two valleys, the superconductor is a non-unitary

**Table 1 | Summary of pairing states in spin-polarized flat bands**

| IR of $D_6$ | $\Delta_{k,\eta} = -\Delta^T_{-k,-\eta}$ | nodes | IR of $C_6$ |
|---|---|---|---|
| $A_1$ | $\sigma_y \chi_{\eta \cdot k} \chi_{C_{2x}k} = -\chi_k$ | ln/pt or ln | $A$ |
| $A_2$ | $\sigma_y \chi_{\eta \cdot k} \chi_{C_{2x}k} = \chi_k$ | ln/n | $A$ |
| $E_2(1,0)$ | $\sigma_y Y_{\eta \cdot k}$ | ln/ln or pt | $E_2(1,0)$ |
| $E_2(0,1)$ | $\sigma_y X_{\eta \cdot k}$ | ln/ln or pt | $E_2(1,0)$ |
| $E_2(1,i)$ | $\sigma_y(X_{\eta \cdot k} + iY_{\eta \cdot k})$ | ln/ln or n | $E_2(1,i)$ |
| $B_1$ | $\eta \hat{\chi}_{\eta \cdot k}, \sigma_z \hat{\chi}_{C_{2x}k} \sigma_z = \hat{\chi}_k$ | n | $B$ |
| $B_2$ | $\eta \hat{\chi}_{\eta \cdot k}, \sigma_z \hat{\chi}_{C_{2x}k} \sigma_z = -\hat{\chi}_k$ | pt | $B$ |
| $E_1(1,0)$ | $\eta \hat{X}_{\eta \cdot k}$ | pt | $E_1(1,0)$ |
| $E_1(0,1)$ | $\eta \hat{Y}_{\eta \cdot k}$ | pt | $E_1(1,0)$ |
| $E_1(1,i)$ | $\eta(\hat{X}_{\eta \cdot k} + i\hat{Y}_{\eta \cdot k})$ | n | $E_1(1,i)$ |

Here $\chi_k$ ($\hat{\chi}_k$) is a real-valued (real and symmetric 2 × 2 matrix-valued) MBZ-periodic function invariant under $C_{3z}$. Furthermore, $X_k$ and $Y_k$ ($\hat{X}_k$ and $\hat{Y}_k$) transform as $x$ and $y$ under $D_3$, generated by $C_{3z}$ and $C_{2x}$, while also being real (and symmetric). The third column indicates the type of nodes—line (ln), point (pt), or none (n)—on a generic Fermi surface for sufficiently small/large order-parameter magnitudes; options separated by "or" indicates that this depends on the normal-state band splitting, see main text. The last column shows which states merge when $D_0 \neq 0$, reducing the point group from $D_6$ to $C_6$.

triplet, while anti-alignment[24,28] leads to a singlet-triplet admixed state[13,27,28]. In both cases, all of the following states are well defined, with the aforementioned spin structures and symmetries given by appropriate combinations of spinless operations and spin rotations (see Supplementary Appendix A1).

We will classify the pairing states according to the irreducible representations (IRs) of the system's point group $D_6$, which is generated by six-fold rotations ($C_{6z}$) along the $z$ axis and two-fold rotation symmetry ($C_{2x}$) along the in-plane $x$ axis. Note a displacement field ($D_0 \neq 0$) breaks the in-plane rotations leading to the point group $C_6$. Importantly, all IRs of $D_6$ and $C_6$ are either even or odd under $C_{2z}$. Choosing the phases of the Bloch states such that $C_{2z}$ acts as $c_{k,\alpha,\eta} \to c_{-k,\alpha,-\eta}$, it holds

$$C_{2z}: \quad \Delta_{k,\eta} \quad \longrightarrow \quad \Delta_{-k,-\eta} = -\Delta^T_{k,\eta}. \quad (2)$$

This immediately implies that the pairing states in all IRs even under $C_{2z}$ ($A_1, A_2, E_2$ of $D_6$) must be anti-symmetric in band space and, thus, entirely band-off-diagonal, whereas the order parameters of the other IRs ($B_1, B_2, E_1$) are symmetric and can contain both band-diagonal and band-off-diagonal components. While superconducting order parameters with finite band-off-diagonal components are rather common in multi-band systems, the existence of pairing states that are constrained to be entirely band-off-diagonal is rather unique and follows from the combination of $C_{2z}$ symmetry and the spin polarization in the normal state. Importantly, this is unaffected by strain or nematic order breaking $C_{3z}$ as long as $C_{2z}$ remains, which guarantees that there are IRs with entirely band-off-diagonal order parameters.

Choosing the phase conventions of the Bloch states such that $C_{2x}$ and $C_{3z}$ act as $c_{k,\alpha,\eta} \to (\sigma_z)_{\alpha\alpha} c_{(k_x,-k_y),\alpha,\eta}$ and $c_{k,\alpha,\eta} \to c_{C_{3z}k,\alpha,\eta}$, respectively, the resulting candidate order parameters are summarized in Table 1. Note that a momentum-independent representation of $C_{2x}$ must be $\sigma_z$ due to the bands' eigenvalues at the Γ-M line, which in turn are connected to the topological obstruction of the flat bands[63]. The reality (Hermiticity) constraint in Table 1 on $\chi$, $X$, and $Y$ ($\hat{\chi}$, $\hat{X}$, and $\hat{Y}$) comes from the residual spinless time-reversal symmetry $\Theta$ of the normal state[64,65]. The two two-dimensional IRs $E_{1,2}$ are each associated with three pairing states—two nematic phases $E_{1,2}(1,0), E_{1,2}(0,1)$ and one chiral state $E_{1,2}(1,i)$.

## Spectral properties

We here have the rather unique situation that there are pairing channels, associated with the IRs $A_{1,2}$ and $E_2$, where the pairing is

constrained by $C_{2z}$ to be entirely band-off-diagonal. One immediate very unusual consequence is that the superconducting order-parameter transforming under the trivial representation ($A_1$) has a symmetry-imposed line of zeros along the Γ-M line, and hence a nodal point in the spectrum. This is related to the topology-induced non-trivial representation of $C_{2x}$ in band space. We refer to ref. 39 for the discussion of other topological nodal points for pairing in obstructed TBG bands. As we will show next, band-off-diagonal pairing leads to additional unusual spectral properties with far-reaching consequences for graphene moiré systems. To this end, consider the following effective Hamiltonian, $\mathcal{H}_{\sigma_y} = \sum_k c^\dagger_{k,\alpha,\eta} c_{k,\alpha,\eta} \xi_{\eta \cdot k,\alpha} + \sum_k [\Delta_k c^\dagger_{k,+} \sigma_y c^\dagger_{-k,-} + \text{H.c.}]$, where the scalar function $\Delta_k$ describes the form of pairing. We will here study two cases that are conventionally considered to be fully gapped, (i) a momentum-independent "s-wave state" ($A_2$ or $A$ pairing in Table 1) where $\Delta_k = \Delta_0$ and (ii) a "chiral p-wave" state, or more precisely an $E_2(1,i)$ state, where $\Delta_k = \Delta_0(X_k + iY_k)$ with $(X_k, Y_k)$ being smooth, MBZ-periodic functions transforming as $(x, y)$ under $C_{3z}$. Furthermore, we parameterize the dispersion, $\xi_{\eta,\alpha}$, of the two flat bands ($\alpha = \pm$) in valley $\eta = \pm$ as $\xi_{k,\alpha} = \epsilon_k - \mu + \alpha \delta_k$, where $\epsilon_k$ and $\delta_k$ are $C_{3z}$ (and, for $D_0 = 0$, $C_{2z}$) symmetric functions.

The Bogoliubov spectrum of $\mathcal{H}_{\sigma_y}$ has four bands, given by $\pm \delta_k \pm \sqrt{(\epsilon_k - \mu)^2 + |\Delta_k|^2}$. Consequently, the excitation gap at momentum $k$ reads as

$$\Delta E_k = \left| |\delta_k| - \sqrt{(\epsilon_k - \mu)^2 + |\Delta_k|^2} \right|, \quad (3)$$

which is shown in Fig. 1d, and therefore exhibits nodes where $|\delta_k| = \sqrt{(\epsilon_k - \mu)^2 + |\Delta_k|^2}$. As long as the band structure has Dirac points, there are points $k_D$ in the Brillouin zone with $\delta_{k_D} = 0$, associated with the blue cross in Fig. 1d. Furthermore, for a metallic normal state, $\mu$ must be within the bandwidth and, hence, there must be a region $R$ in momentum space where $|\delta_k| > |\epsilon_k - \mu|$. For the momentum-independent $A_2$ state, $\Delta_k = \Delta_0$, this implies that there exists $\Delta_0^c > 0$ such that there is $k^* \in R$ with parameters (such as the blue circle) above the red solid line in Fig. 1d as long as $|\Delta_0| < \Delta_0^c$. By continuity, this means that there must be a nodal point on any line connecting $k_D$ and $k^*$. Consequently, for $\mu$ within the bandwidth and $\delta_{k_D} = 0$ for some $k_D$, the $A_2$ will always have a nodal line if $|\Delta_0|$ is sufficiently small, consistent with the intuitive picture based on the Bogoliubov spectrum in Fig. 1a–c.

We illustrate this further in Fig. 1e using a toy model with $\delta_k = t|1 + e^{ia_1 \cdot k} + e^{-ia_2 \cdot k}|$ and $\epsilon_k = t' \sum_{j=1}^3 \cos a_j \cdot k$, $a_j = [C_{3z}]^{j-1}(\sqrt{3},0)^T$. This leads to the second unexpected conclusion that, for any pairing mechanism, including conventional electron–phonon coupling, the leading instability either has nodal lines in a finite region below $T_c$ or transforms non-trivially under the symmetries of the normal state. For electron–phonon pairing (or pairing mediated by the fluctuations of any time-reversal-symmetric order parameter[52], such as the T-IVC state) this is particularly unexpected since it is generally believed to always lead to a fully gapped state that transforms trivially under all symmetries. In fact, this can be proven in general terms[51,52], even for spin-orbit-split Fermi surfaces and beyond mean-field theory[52]. The crucial difference to these works, however, is that spinful time-reversal is broken in our case such that the Fermi-Dirac constraint is inconsistent with a non-sign-changing, band-diagonal pairing state. This leads to the unique situation that although electron–phonon coupling will lead to entirely attractive interactions in the Cooper channel, the superconducting energetics is frustrated: the dominant pairing state is determined by whether the energetic loss due to non-resonant band-off-diagonal Cooper pairs ($A_2$ pairing) or the costs from sign changes of the order parameter (such as $B_1$) are less harmful. We

will demonstrate this explicitly by a model calculation in sec. "electron–phonon coupling below, where either $A_2$ or $B_1$ is dominant, depending on the form of the electron–phonon coupling".

Let us first, however, discuss the general spectral properties of the "chiral $p$-wave" state which is canonically expected to be fully gapped as long as the Fermi surfaces do not cross the zeros of $X_{\boldsymbol{k}} + iY_{\boldsymbol{k}}$. Three of these zeros have to be at the $\Gamma$, $K$, and $K'$ points as a consequence of $C_{3z}$ symmetry. In the absence of fine-tuning, $X_{\boldsymbol{k}} + iY_{\boldsymbol{k}}$ will have vortices at these points with vorticity $v = +1$. As can be seen in Fig. 1f, where we show the phase of $X_{\boldsymbol{k}} + iY_{\boldsymbol{k}}$ using an admixture of the two lowest-order terms, the net vorticity of $+3$ at these high-symmetry points has to be compensated by anti-vortices at generic momenta. The lowest possible number is three $C_{3z}$-related vortices, which appear near the M points in Fig. 1f. If it holds $|\delta_{\boldsymbol{k}}| > |\epsilon_{\boldsymbol{k}} - \mu|$ at any of these zeros $\boldsymbol{k} = \boldsymbol{k}_j$, we obtain a point above the red line in Fig. 1d and, thus, a nodal point along any contour between that $\boldsymbol{k}_j$ and $\boldsymbol{k}_D$; as opposed to the $A_2$ state, this holds irrespective of the value of $\Delta_0$ and therefore all the way to zero temperature. In summary, we find that also the $E_2(1, i)$ "chiral $p$-wave" state is not generically fully gapped but instead will exhibit a nodal line encircling any zero $\boldsymbol{k}_j$ of $X_{\boldsymbol{k}} + iY_{\boldsymbol{k}}$ with $|\delta_{\boldsymbol{k}_j}| > |\epsilon_{\boldsymbol{k}_j} - \mu|$. This leads to an interesting filling dependence of the superconducting gap, as we illustrate in our toy model in Fig. 1g along with the criterion $D_{\boldsymbol{k}_j}$ : $= |\delta_{\boldsymbol{k}_j}| - |\epsilon_{\boldsymbol{k}_j} - \mu| > 0$ evaluated at the vortices at $\Gamma$, K/K', and near M. Depending on $\mu$, $D_{\boldsymbol{k}}$ is positive only near the $\Gamma$ point or only in a region surrounding the vortices close to the M points, leading to nodal lines encircling $\Gamma$ and near the M points, respectively, as shown in the inset of Fig. 1g. These regimes are separated by a fully gapped region where $D_{\boldsymbol{k}} < 0$ for all $\boldsymbol{k}$, which could explain the fully gapped to nodal transition seen in tunneling experiments[49] when the filling fraction is changed. Note that $D_{\boldsymbol{k}_j} = -|\epsilon_{\boldsymbol{k}_j} - \mu| \le 0$ for $\boldsymbol{k}_j$ at the K and K' points. In Fig. 1g, $D_K = D_{K'}$ vanishes close to the top of the band, which simply means that the Fermi surfaces cross the K, K' points and the superconductor has nodal points for this fine-tuned value of the chemical potential.

## Fluctuation-induced pairing

Having discussed the unique energetics of pairing and spectral properties of the resulting superconductors in spin-polarized quasi-flat-bands with Dirac cones on a general level, we next study these aspects more explicitly by solving the superconducting self-consistency equations in the flat bands common to alternating-twist graphene systems. We will start with pairing induced by fluctuations of a nearby symmetry-broken phase. To this end, we will couple the low-energy electrons introduced in Eq. (1) to a collective bosonic field $\phi_j(\boldsymbol{q}) = \phi_j^\dagger(-\boldsymbol{q})$ via

$$\mathcal{H}_\phi = \sum_{\boldsymbol{k},\boldsymbol{q},j} c^\dagger_{\boldsymbol{k}+\boldsymbol{q},\alpha,\eta} \lambda^j_{\alpha,\alpha';\eta,\eta'} c_{\boldsymbol{k},\alpha',\eta'} \phi_j(\boldsymbol{q}), \qquad (4)$$

where the Hermitian matrices $\lambda^j$ capture the nature of the correlated insulating phase; we here choose and normalize $\lambda^j$ such that $(\lambda^j)^2 = \mathbb{1}$. Both for twisted bi-[9] and trilayer graphene[14,15,29], the stable phases emerging out of the $U(4) \times U(4)$[9] manifold in the chiral-flat (decoupled) limit are natural candidates. Integrating out the bosonic modes, we obtain an effective electronic interaction which in the for superconductivity relevant intervalley Cooper channel reads as

$$\mathcal{H}^\phi_{\text{int}} = -\sum_{\boldsymbol{k},\boldsymbol{k}'} \chi_{\boldsymbol{k}-\boldsymbol{k}'} \mathcal{V}_{(\eta,\alpha,\beta),(\eta',\alpha',\beta')}$$
$$\times c^\dagger_{-\boldsymbol{k},\beta,-\eta} c^\dagger_{\boldsymbol{k},\alpha,\eta} c_{\boldsymbol{k}',\alpha',\eta'} c_{-\boldsymbol{k}',\beta',-\eta'}, \qquad (5)$$

with vertex

$$\mathcal{V}_{(\eta,\alpha,\beta),(\eta',\alpha',\beta')} = t_\phi \sum_j \left[\lambda^j_{\beta,\eta;\beta',\eta'}\right]^* \lambda^j_{\alpha,\eta;\alpha',\eta'}, \qquad (6)$$

$t_\phi = \pm 1$ encoding whether the order parameter is even or odd under time-reversal, $\Theta\phi_j(\boldsymbol{q})\Theta^\dagger = t_\phi\phi_j(\boldsymbol{q})$, and $\chi_{\boldsymbol{q}} > 0$ denoting the (static) susceptibility of $\phi_j$.

Before discussing numerical results for the full model, we first focus on perfectly flat bands. In this limit, the leading superconducting instability within mean-field theory is given by the largest eigenvalue of $\mathcal{V}$ in Eq. (6) viewed as a matrix in the multi-index ($\eta$, $\alpha$, $\beta$). Furthermore, if there is an anti-symmetric, valley-off-diagonal matrix $D$ obeying (see Methods)

$$[D\eta_x, \lambda^j]_{-t_\phi} \equiv D\eta_x\lambda^j - t_\phi\lambda^j D\eta_x = 0, \qquad (7)$$

the associated leading superconducting order parameter in Eq. (1) is given by $(\Delta_{\boldsymbol{k},\eta})_{\alpha,\alpha'} = \delta_{\boldsymbol{k}}(D\eta_x)_{\alpha,\eta;\alpha'\eta}$ with $\delta_{\boldsymbol{k}} > 0$; here $\eta_j$ denotes Pauli matrices in valley space and the precise form of $\delta_{\boldsymbol{k}}$ is determined by $\chi(\boldsymbol{q})$.

## T-IVC fluctuations

Motivated by recent experiments[7] providing direct evidence for T-IVC order, we start with T-IVC fluctuations as a pairing glue. In the $U(4) \times U(4)$ symmetric limit, the T-IVC state is associated with $\lambda^j = \sigma_0\eta_j, j = x, y$, within our conventions. Since $t_\phi = +1$, we are looking for $D\eta_x$ that commutes with $\lambda^j$. Interestingly, there is a unique anti-symmetric, valley-off-diagonal matrix $D \propto \sigma_y\eta_x$ with that property, implying that the leading pairing state has the form $\Delta_{\boldsymbol{k},\eta} = \sigma_y\delta_{\boldsymbol{k}}, \delta_{\boldsymbol{k}} > 0$. This is exactly the $A_2$ state in Table 1, which, as discussed above, will have nodal lines at least in the vicinity of $T_c$ when a finite band dispersion is taken into account. Intuitively, the fact that $A_2$ pairing is favored can be understood by noticing that the valley-off-diagonal form of $\lambda^j$ leads to an attractive interaction across the valleys, which penalizes the $B_1$ state with its sign change between the two valleys. In fact, it holds $\mathcal{V}_{(\eta,\alpha,\beta),(\eta',\alpha',\beta')} = (1 - \eta\eta')\sum_{\mu=0}^3 (\sigma_\mu^*)_{\alpha,\beta}(\sigma_\mu)_{\alpha',\beta'}$ showing explicitly that it is repulsive (attractive) in the $B_1$ ($A_2$) channel.

To go beyond the flat-band limit, we solve the superconducting mean-field equations numerically. We take the flat TBG bands from the continuum model[66] as the starting point. To capture the spin-polarized normal state, we supplement it with Coulomb repulsion and perform HF calculation (see Supplementary Appendix A for details). As can be seen in the resulting band structure shown in Fig. 2a with interaction renormalization assuming filling fraction $v = 2$, this not only pushes one spin flavor below the Fermi level but also induces significant band renormalizations. For our subsequent study of superconductivity, we project onto the two bands at the Fermi level and associate them with the creation operators $c_{\boldsymbol{k},\alpha}$ in the interactions in Eqs. (4) and (5). In our numerical computations, we choose $\chi(\boldsymbol{q}) = \frac{1}{A_m}\frac{V}{\alpha^2 + |\boldsymbol{q}|^2/k_\theta^2}$ where $A_m$ is the real space area of a moiré unit cell, and take $\alpha = 0.05$ for concreteness, although we checked our main conclusion do not crucially depend on this form. In all of our numerics, we work at doping $v = 2.5$.

As expected, we indeed find that the $A_2$ state dominates, both right at the critical temperature $T_c$, obtained from the linearized gap equation, and at $T = 0$ as we show by iteratively solving the full self-consistency equation (see Supplementary Appendix C). One crucial effect of the finite dispersion and splitting between the bands is that a finite interaction strength, $V > V_{c,1}$, is required to stabilize the superconducting phase, as can be seen in the plot of $T_c$ in Fig. 2b. Superconductivity ceases to be a weak-coupling instability as the Bloch states $(\boldsymbol{k}, \alpha, \eta)$ and $(-\boldsymbol{k}, \alpha', -\eta)$ are not degenerate for $\alpha \ne \alpha'$, cutting off the logarithmic divergence known from BCS theory. The quasiparticle spectrum and order parameter of superconductivity from $T = 0$ numerics are shown in Fig. 2c, d. In accordance with our general discussion above, we observe that the order parameter only has finite components proportional to $\sigma_y$, which do not mix with the band-even contributions $\propto \sigma_{0,x,z}$ as a result of $C_{2z}$ symmetry. Furthermore, it does not change sign as a function of $\boldsymbol{k}$ and, for sufficiently small $V$ but still

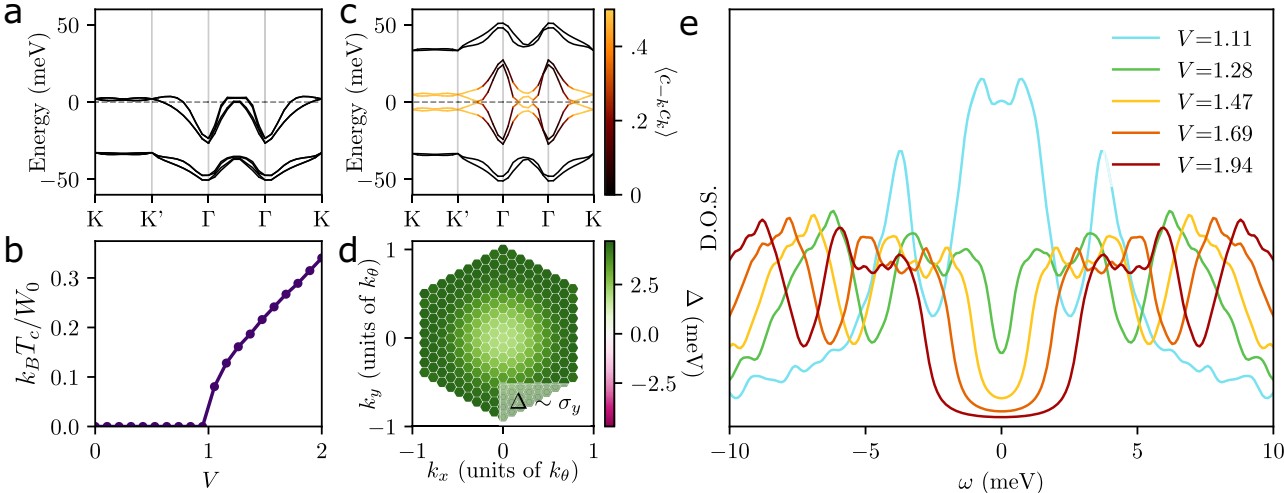

**Fig. 2 | Pairing mediated by T-IVC fluctuations.** We show **a** the band structure of the normal state with spin polarization (K, K', and Γ label the high-symmetry points of the moiré scale Brillouin zone) and **b** the critical temperature $T_c$ (in units of the maximum band splitting $W_0 \simeq 9.4$ meV) as a function of coupling strength $V$ measured in units of the critical coupling $V_{c,1} = 105$ meV·nm$^2$ obtained from the linearized gap equation. The band structure (with color indicating the band-projected value of the anomalous correlator) of the $A_2$ state and its order parameter are shown in **c** and **d**. The DOS of the $T = 0$ superconductor for several different values of coupling strength $V$ is plotted in **e**. The DOS was computed as $\sum_{\mathbf{k}} \delta(E_{\mathbf{k}} - \omega)$, replacing the $\delta$ function with Lorentzians with half width at half max 0.3 meV (much smaller than the typical superconducting order parameter). The critical coupling $V_{c,2}$ where the nodal lines disappear is $V_{c,2} \simeq 1.4 V_{c,1}$.

with $V > V_{c,1}$, the nodal lines in the superconducting spectrum persist all the way to $T = 0$, while the nodal line is gapped out at low $T < T_c$ if $V > V_{c,2}$.

The interaction-strength-dependence of the superconducting gap can be more clearly seen in Fig. 2e, where we show the DOS for the self-consistent solution at $T = 0$. For large $V$, the superconductor becomes fully gapped at $T = 0$, leading to a $U$-shaped DOS. With smaller $V$, the magnitude of the order-parameter decreases and the superconductor eventually exhibits nodal lines, as explained above. In the regime just before these nodal lines appear, there is an increase in the DOS near the Fermi level, roughly when the order parameter and the maximal band splitting are comparable, leading to a V-shaped DOS (green line). The lifetime parameter used to compute the DOS is 0.3 meV; this choice was based on our $k$-grid spacing. While it is not necessarily small with respect to the tunneling gap (which vanishes at $V_{c,2}$), it is small with respect to $\Delta(\mathbf{k})$, which is of order 5 meV just as the state is becoming fully gapped for our choice of normal state. This behavior of the DOS with interaction strength may offer a natural explanation for the $U$-shaped tunneling conductance measurements near $\nu = 2$ and V-shaped tunneling conductance measurements near $\nu = 3$ observed in TTG[49]; if we are considering T-IVC fluctuations of the insulator at $\nu = 2$, then it may be reasonable to expect the coupling to these fluctuations could grow weaker as we dope towards $\nu = 3$, in line with the experimentally observed $\nu$ dependence.

Note that the regime we call V-shaped here is strictly speaking fully gapped. However, the crucial difference to the BCS state is that the gap is much smaller than the order-parameter magnitude as a result of the different Bogoliubov spectrum in Eq. (3). This is why, depending not only on the magnitude of the pairing but also on the precise form of the normal state, the resulting tunneling spectra can resemble those observed experimentally[48,49], such as the green curve in Fig. 2e, making the $A_2$ state an attractive candidate. The regime of small $V$ where stable superconductivity with true Bogoliubov Fermi surfaces is observed can further exhibit a peak at $\omega = 0$ which is due to a Van Hove singularity crossing the Fermi level, see blue curve in Fig. 2e; while this peak has not been observed experimentally, its presence crucially depends on details of the normal-state band structure and is only found to be energetically favored in a very small regime of $V$ in our model.

## Electron–phonon coupling

To illustrate that the off-diagonal $A_2$ state is more generally favored beyond just T-IVC fluctuations, we next discuss electron–phonon coupling, which is frequently considered a plausible pairing mechanism for twisted moiré systems[33,35–39]. Similar to ref. 35, we use that the optical $A_1$, $B_1$, and $E_2$ phonon modes are known[67] to dominate the electron–phonon coupling in single-layer graphene. As these are optical phonons, we further assume that the impact of the interlayer coupling on the phonons can be neglected and arrive at

$$\mathcal{H}_{EP} = \int d\mathbf{r}\, \psi_{\ell,s}^\dagger(\mathbf{r}) \Big[ g_{A_1} \Lambda_{A_1} u_{A_1,\mu}(\mathbf{r}) + g_{B_1} \Lambda_{B_1} u_{B_1,\mu}(\mathbf{r}) + g_{E_2} \boldsymbol{\Lambda}_{E_2} \cdot \mathbf{u}_{E_2,\mu}(\mathbf{r}) \Big] (\boldsymbol{v}_\mu)_\ell \psi_{\ell,s}(\mathbf{r}) \tag{8}$$

for the electron–phonon coupling, where $\boldsymbol{v}_\mu$ encode the layer structure of the modes (see Methods). Symmetry dictates that the vertices $\Lambda_g$ are given by $\Lambda_{A_1} = \eta_x \rho_x$, $\Lambda_{B_1} = \eta_y \rho_x$, and $\boldsymbol{\Lambda}_{E_2} = (\eta_z \rho_y, -\rho_x)$ where $\rho$ acts on the microscopic sublattice basis. Integrating out the phonons and projecting to the flat bands, we obtain an effective electron-electron interaction (see Methods)

$$\mathcal{H}_{\text{int}}^C = -\sum_{\mathbf{k},\mathbf{k}'} V_g \Big[ \lambda_{\mathbf{k},\beta,\eta;\mathbf{k}',\beta',\eta'}^{g,j,\mu} \Big]^* \lambda_{\mathbf{k},\alpha,\eta;\mathbf{k}',\alpha',\eta'}^{g,j,\mu} \times c_{-\mathbf{k},\beta,-\eta}^\dagger c_{\mathbf{k},\alpha,\eta}^\dagger c_{\mathbf{k}',\alpha',\eta'} c_{-\mathbf{k}',\beta',-\eta'}, \tag{9}$$

where the coupling constants $V_g$ of the three different phonon modes $g = A_1, B_1, E_2$ are estimated to obey $V_{A_1} = V_{B_1} \simeq 1.33 V_{E_2}$ for parallel spins in the two valleys, while $V_{A_1} = V_{B_1} = 0$ for anti-parallel spins. From Eq. (9), it is clear that the induced interaction would be always completely attractive if we focused on intra-band pairing, $\alpha = \alpha' = \beta = \beta'$, which in spinful systems generically favors the trivial pairing channel[51,52]. In our case, the combination of two energetically close bands and the trivial pairing being purely band-off-diagonal leads to competition between different superconductors, even with electron–phonon coupling alone.

To demonstrate this, we study intra-valley pairing within the mean-field approximation and parametrize the relative strength of the different phonon modes with an angle variable $\theta_{ph}$ according to

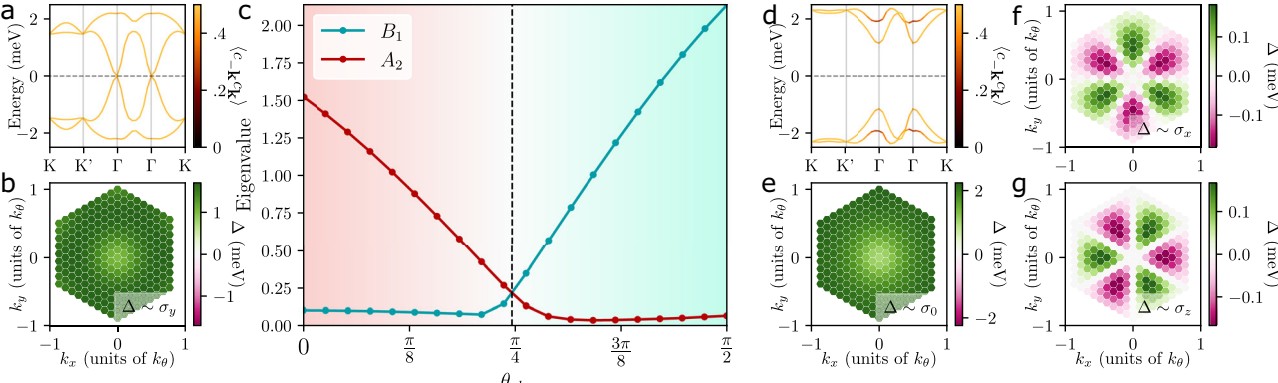

**Fig. 3 | Pairing from electron–phonon coupling.** We show **a** the band structure and **b** the self-consistent order parameter of the $A_2$ pairing for $\theta_{\text{ph}} = 0$ and $T = 0$. The eigenvalues corresponding to the $A_2$ and $B_1$ pairings in the linearized gap equation at $T = 5$ K, which is close to their $T_c$, are shown in **c** as a function of $\theta_{\text{ph}}$. We show an example of the band structure (**d**) of the $B_1$ pairing and its order parameter (**e**, **f**, **g**). In accordance with symmetry, the $A_2$ ($B_1$) state only has order-parameter components $\propto \sigma_y$ ($\propto \sigma_{0,x,z}$). We took $\nu = 2.5$ and $V_0 = 250$ meV $\cdot$ (nm)$^2$ with a continuum-model bandwidth $\simeq 2$ meV. We point out that if $A_1$ phonons are dominant, as suggested by recent experimental work[61] and past theoretical study in mono-layer graphene[68], we would expect our $A_2$ pairing to dominate assuming the pairing potential is sufficiently large. We also emphasize that although the pairing functions for $A_2$ pairing (**b**) when $\theta = 0$ and $B_1$ pairing when $\theta = \pi/2$ (**e**, **f**, **g**) are roughly equal, the excitation spectra show the $B_1$ state with a band gap on the order of the pairing strength (**d**) while the $A_1$ state's band gap is nearly zero, see **a**.

$V_{A_1} = V_{B_1} = V_0 \cos \theta_{\text{ph}}, V_{E_2} = V_0 \sin \theta_{\text{ph}}$. The results of the mean-field calculation are summarized in Fig. 3. We see that the $A_2$ pairing state is favored by the intervalley phonons ($\theta_{\text{ph}} = 0$) inspite of its band-off-diagonal nature leading to a suppressed gap [see Fig. 3a]. This is natural as these phonons mediate an attractive interaction between the two valleys which disfavors the $B_1$ state, similar to T-IVC fluctuations. In fact, focusing on the leading, momentum-independent term, $\lambda^{g,+}_{\boldsymbol{k},\boldsymbol{k}'} \rightarrow \lambda^{g,+}, g = A_1, B_1$, symmetry dictates $\lambda^{A_1,+} \propto \sigma_0 \eta_1$ and $\lambda^{B_1,+} \propto \sigma_0 \eta_2$ in the chiral limit (see Supplementary Appendix D3). This maps the problem exactly to that of T-IVC fluctuations, immediately explaining why the order parameter has a fixed sign in Fig. 3b. As $\theta_{\text{ph}}$ is increased, the $B_1$ state is favored (roughly for $\theta_{\text{ph}} > \pi/4$) as can be seen in Fig. 3c. This is expected since the intra-valley $E_2$ phonon mediates an attractive interaction within each valley such that the energy gain due to the enhanced gap [Fig. 3d], associated with the band-diagonal matrix elements of the $B_1$ state, will overcompensate the energetic loss due to the sign change of $B_1$'s order-parameter between the two valleys. This picture is consistent with the dominant and non-sign-changing nature of the band-diagonal components of the $B_1$ state, see Fig. 3e–g. Finally, this behavior can also be understood by applying the commutator criterion in Eq. (7) in the microscopic sublattice basis, see Supplementary Appendix D1.

This shows that, as opposed to the conventional scenario[51,52], there are two possible leading superconducting states and the superconducting pairing state does not transform trivially under the symmetries of the system even when phonons alone provide the pairing glue. We have checked in our $T = 0$ numerics that a 60–70 meV $\cdot$ (nm)$^2$ coupling to $A_1$ and $B_1$ phonons (based on ref. [67]) is roughly of the order needed to stabilize the $A_2$ pairing, assuming the normal state is the flat bands of the un-renormalized continuum model, which in our case has a bandwidth of 2 meV. However, we note that if the interaction-renormalized band splitting is much larger than the continuum-model bandwidth, or if the normal state has anti-parallel spins in either valley, additional particle-hole fluctuations, such as those of T-IVC order, will also be required for pairing. An interesting scenario arises for anti-parallel spins in the two valleys as a magnetic field will cant the spins and, hence, increase the projection of the intervalley phonon matrix elements to the flat bands. At least in TTG, with the suppressed orbital coupling, this could give rise to re-entrant superconductivity at high fields[25].

## Other particle-hole fluctuations

Finally, we discuss pairing induced by fluctuations of other particle-hole instabilities. In Table 2, we list the resulting leading superconductors taking $\lambda^j$ in Eq. (4) to be any of the different strong-coupling candidate order parameters[9,13–15,29]. In particular, in addition to the T-IVC, we will consider the time-reversal-odd Kramers intervalley coherent state (K-IVC), and time-reversal-odd and -even sublattice polarized states (SLP− and SLP+). To analyze how sensitive our conclusions are to the precise form of the coupling of the strong-coupling fluctuating orders to the electrons, we also perform numerics by projecting momentum-independent coupling vertices in the microscopic basis with the correct symmetries (see, e.g., Table II in ref. [13]), listed as $\bar{\lambda}^j$ in Table 2, to the flat bands. In the band basis, this leads to momentum-dependent coupling vertices, cf. Eq. (9). Motivated by recent experiments[7], we will also consider fluctuations of an additional nematic, time-reversal symmetric, layer-odd, intervalley coherent state (N-IVC)[62] which is not a candidate ground state in the strong-coupling limit; unlike the other strong-coupling ground states, the N-IVC has no momentum-independent representation in the flat-band basis but does have a momentum-independent matrix order parameter in the sublattice basis which takes the form $\lambda^{(jj')} = (\eta_x, \eta_y)_j (\rho_0, \rho_z)_{j'}$. The results for fluctuations of the projected strong-coupling orders $\bar{\lambda}^j$ in Table 2 and of the projected N-IVC state are shown in Fig. 4, where we use the angle $\theta_{\text{fluc}}$ to tune the relative strength between T-IVC and any of the other type of fluctuation-induced interactions by multiplying the T-IVC interaction potential

**Table 2 | Leading superconducting states in the flat-band limit, following from Eq. (7), for pairing mediated by fluctuations of the indicated orders, defined by using $\lambda^j$ in Eq. (4)**

| Fluctuating order | | | Leading superconductor | |
|---|---|---|---|---|
| type | $\lambda^j$ | $\bar{\lambda}^j$ | $\Delta_{\boldsymbol{k},\eta}$ | IR |
| T-IVC | $\sigma_0 \eta_{x,y}$ | $\rho_x \eta_{x,y}$ | $\sigma_y \delta_{\boldsymbol{k}}$ | $A_2$ |
| K-IVC | $\sigma_y \eta_{x,y}$ | $\rho_y \eta_{x,y}$ | $\sigma_0 \eta \delta_{\boldsymbol{k}}$ | $B_1$ |
| SLP + | $\sigma_y \eta_z$ | $\rho_z \eta_0$ | $\sigma_y \delta_{\boldsymbol{k}}, \sigma_0 \eta \delta_{\boldsymbol{k}}$ | $A_2, B_1$ |
| SLP − | $\sigma_y \eta_0$ | $\rho_z \eta_z$ | $\sigma_x \eta \delta_{\boldsymbol{k}}, \sigma_z \eta \delta_{\boldsymbol{k}}$ | $B_2, B_1$ |

Here, $\delta_{\boldsymbol{k}} > 0$ and states separated by commas are degenerate. The couplings in the microscopic basis, used in Fig. 4 for the respective orders, are listed under $\bar{\lambda}^j$. Except for SLP−, the leading superconducting states for $\lambda^j$ and $\bar{\lambda}^j$ are the same (cf. Fig. 4 and Supplementary Appendix D1).

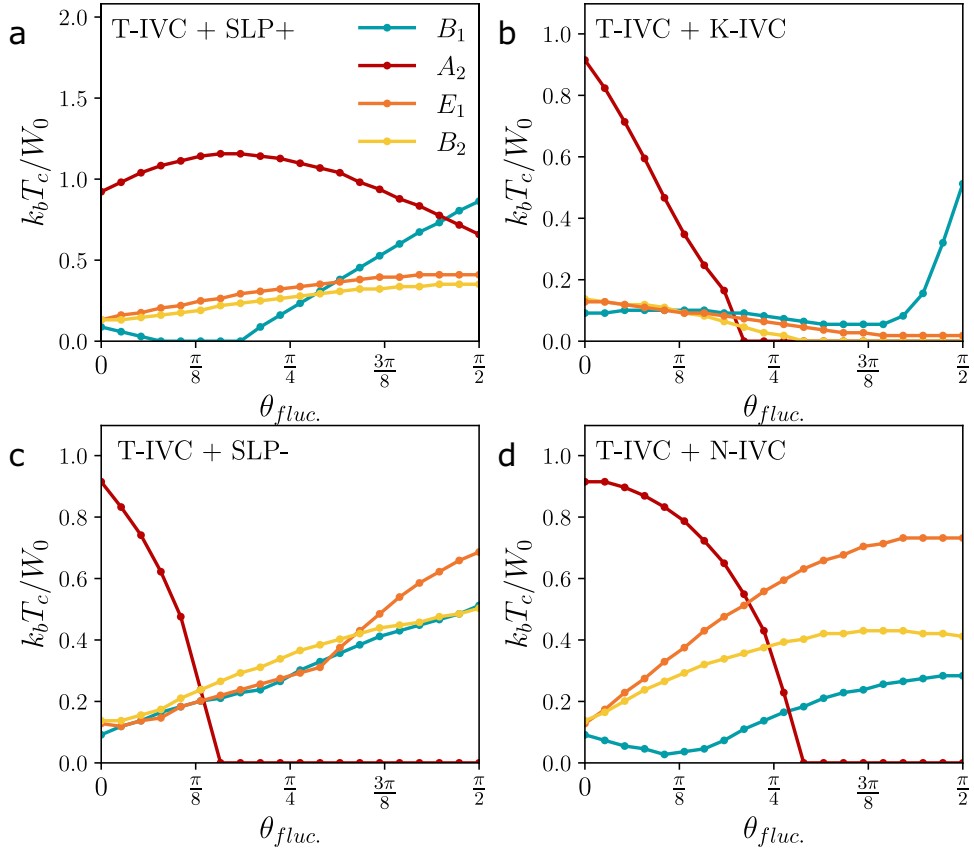

**Fig. 4 | Pairing for different particle-hole fluctuations.** These are defined by the coupling matrices $\bar{\lambda}^j$ listed in Table 2. Similar to Fig. 3c, we show $T_c$ of the leading pairing states, where $\theta_{\text{fluc.}}$ tunes the relative strength between T-IVC-induced interactions ($\propto \cos\theta_{\text{fluc.}}$) and interactions ($\propto \sin\theta_{\text{fluc.}}$) coming from fluctuations of **a** SLP+, **b** K-IVC, **c** SLP−, and **d** N-IVC fluctuations.

with $\cos(\theta_{\text{fluc.}})$ and the other fluctuation potential with $\sin(\theta_{\text{fluc.}})$. In our microscopic numerics, we have taken a potential form $\chi(\boldsymbol{q}) = \frac{1}{A_m} \frac{V}{\alpha^2 + |\boldsymbol{q}|^2/k_\theta^2}$ again with $\alpha = 0.2$ and with $V = 4200 \, \text{meV} \cdot (\text{nm})^2$. We chose the value of $V$ such that the transitions between the different pairing states are clearly visible in Fig. 4 when varying $\theta_{\text{fluc.}}$. In accordance with the prediction for $\bar{\lambda}^j$ in Table 2, SLP + fluctuations further stabilize the $A_2$ superconductor, see Fig. 4a. As such, the band-diagonal $B_1$ superconducting channel, where SLP + fluctuations are also attractive, can become the leading channel (favored over $A_2$ as a result of the finite bandwidth) only very close to $\theta_{\text{fluc.}} = \pi/2$. K-IVC fluctuations, however, are repulsive for $A_2$ pairing and favor the $B_1$ state more strongly.

So far, the strong-coupling ($\lambda$) and sublattice ($\bar{\lambda}^j$) form of the couplings in Table 2 lead to the same conclusions. This is different for SLP− fluctuations (Fig. 4d), where the projection-induced momentum dependence in the band basis can stabilize the $E_1$ superconductor. This can be understood by applying Eq. (7) in a sublattice basis (see Supplementary Appendix D1). We also find the $E_1$ state when fluctuations of the N-IVC state of ref. 62 dominate. Examples of the $E_1$ nematic and $B_2$ order parameters that emerge for SLP− fluctuations or N-IVC fluctuations are shown in Supplementary Appendix F. We point out that the nematic $E_1$ pairing is also an interesting candidate given that despite having nonzero pairing in the $\sigma_0$, $\sigma_x$, $\sigma_z$ channels, it will be nodal as long as the $\sigma_x$ components do not gap out the nodes in the band-diagonal parts.

## Discussion

Taken together, we see that the proposed band-off-diagonal $A_2$ superconductor is an especially attractive candidate for TBG and TTG:

first, it can lead to both V-shaped or U-shaped DOS, depending on lifetime parameters, the normal state, and the coupling strength $V$, see Fig. 2e. As these parameters might vary from sample to sample and within a sample (e.g., $V$ is expected to decrease upon doping further away from the insulator), this can naturally explain the tunneling data of[48,49]. We emphasize however that at least at the level of our mean-field numerics, we only expect a V-shape in the regime where the superconducting pairing is of the order of the bandwidth; this is the regime, where although the pairing is finite and can be quite large, the gap in the superconducting spectrum is either just closing or very small relative to the pairing. Increasing the pairing further will lead to an evolution from V to U-shaped while decreasing the pairing will eventually lead to a nodal Fermi surface and presumably a peak at zero energy in the DOS. Second, despite its interband nature, $A_2$ is the unique pairing state that is favored by fluctuations of two out of the four strong-coupling candidates we consider for the correlated insulator, see Fig. 4a–c. What is more, this includes the T-IVC state, signatures of which are observed in recent experiments[7]. Finally, it is also favored by the likely dominant[61,68] optical intervalley phonon modes. We emphasize that, both in the case of fluctuating correlated insulators and phonons, the minimum attractive coupling needed to stabilize a purely band-off-diagonal state depends on the energy splitting between the two flat bands in the normal state; if the bands of our normal state are closer to degenerate, irrespective of the total bandwidth, the needed coupling to stabilize the $A_2$ pairing in mean-field will decrease.

The other band-off-diagonal superconductor we identify transforms under the IR $E_2$, i.e., can be thought of as a $p$-wave state. Its spectral properties also agree well with the experiment as the chiral configurations, $E_2(1, i)$, which is favored within mean-field theory over a

nematic $E_2$ state, can also have nodal regions, depending on filling. As can be seen in Fig. 1g, this can lead to a transition from gapped to nodal when increasing the electron filling starting at $\nu \simeq 2$. However, as opposed to the $A_2$ state, $E_2$ does not naturally appear as leading instability when considering optical phonons or fluctuations of any of the strong-coupling order parameters of the correlated insulator. While this makes it energetically less natural than $A_2$, we cannot exclude it since its phenomenology agrees well with the experiment and since the precise form of the coupling of the dominant low-energy collective excitations are not known—significant momentum dependencies beyond $\lambda^j$ and $\bar{\lambda}^j$ in Table 2 could stabilize $E_2$ pairing as well. We also find in our numerics a nematic $E_1$ state which may be preferred over its chiral version in the presence of sufficient strain or due to fluctuation corrections[27,53–55]. We find the $E_1$ state is the leading instability of nematic IVC fluctuations and SLP− fluctuations, and is a subleading instability of T-IVC fluctuations. The $E_1$ state is interesting in its own right, as it can also be nodal.

As superconductivity might further coexist with T-IVC order[7], we have checked (see Supplementary Appendix E) that this does not alter our main observation: the preserved $C_{2z}$ symmetry still allows for entirely band-off-diagonal states, with transitions from nodal to full gapped, which are stabilized (among other fluctuations) by intervalley phonons.

For the future, it will be interesting to go beyond mean-field and analyze the competition of our band-off-diagonal states with odd-frequency pairing, which we study in a follow-up work[69]. It also seems promising to study Andreev reflection[48,49] for our interband pairing scenario. On a more general level, our work shows that the observation of nodal pairing in twisted graphene systems does not immediately exclude a chiral superconducting state nor an entire electron–phonon-based pairing mechanism. It illustrates that a microscopic understanding of the superconducting states in graphene moiré systems requires taking into account their intrinsically multi-band nature.

*Note added*. Just before posting our work, ref. 70 appeared online, which discusses pairing induced by $A_1$ phonons in spinful TBG bands.

## Methods
### Flat-band limit
To derive Eq. (7), we take the flat-band limit, $\xi_{k,\pm} \to 0$, in the linearized gap equation. For the interaction defined in Eqs. (4)–(6), we get (with moiré cell area $A_m$)

$$(\Delta_{k,\eta})_{\beta,\beta'} = t_\phi \frac{1}{4 A_m T} \sum_{k'} \chi_{k-k'}$$
$$\times \sum_j \left[\lambda^j_{\beta',\eta;\alpha',\eta'}\right]^* \lambda^j_{\beta,\eta;\alpha,\eta'} (\Delta_{k',\eta'})_{\alpha,\alpha'}. \qquad (10)$$

We define $(\hat{\Delta}_k)_{\alpha,\eta;\alpha',\eta'} := (\Delta_{k,\eta})_{\alpha,\alpha'} \delta_{\eta,\eta'}$ and note that finding the leading superconducting state according to Eq. (10) is equivalent to determining $\hat{\Delta}_k$ that maximizes the functional

$$\mathcal{F}[\hat{\Delta}_k] := \frac{\sum_{k,k',j} \chi_{k-k'} t_\phi \, \mathrm{tr}[\lambda^j \hat{\Delta}_{k'} (\lambda^j)^\dagger \hat{\Delta}_k^\dagger]}{\sum_k \mathrm{tr}[\hat{\Delta}_k^\dagger \hat{\Delta}_k]}. \qquad (11)$$

Since $\chi_{k-k'} > 0$, the maximum value will be reached if we can maximize $t_\phi \, \mathrm{tr}[\lambda^j \hat{\Delta}_{k'} (\lambda^j)^\dagger \hat{\Delta}_k^\dagger]$ for each $k, k', j$ separately. As the Frobenius inner product $\langle A, B \rangle_F = \mathrm{tr}[A^\dagger B]$ reaches its maximum (minimum) at fixed $\langle A, A \rangle$ and $\langle B, B \rangle$, if $A = cB$ with $c > 0$ ($c < 0$), $t_\phi \, \mathrm{tr}[\lambda^j \hat{\Delta}_{k'} (\lambda^j)^\dagger \hat{\Delta}_k^\dagger]$ is maximized if $\hat{\Delta}_k = t_\phi c_{k,k'} \lambda^j \hat{\Delta}_{k'} (\lambda^j)^\dagger$ with $c_{k,k'} > 0$. For the ansatz $\hat{\Delta}_k = \delta_k D \eta_x$ (and *assuming* for now that $\delta_k$ has a fixed sign for all $k$), this

is obeyed if

$$D \eta_x = t_\phi \lambda^j D \eta_x (\lambda^j)^\dagger, \quad \forall j. \qquad (12)$$

We state Eq. (12) as the (anti)commutator condition (7) in the main text [equivalent if $(\lambda^j)^2 = \mathbb{1}$], not only because it highlights the simple algebraic and basis independent nature of the condition but also since it emphasizes the similarities to the generalized Anderson theorem of[71,72].

If we can find a solution to Eq. (12), we know that the maximum (or at least one of the possibly degenerate maxima) of $\mathcal{F}[\hat{\Delta}_k]$ is of the form of $\hat{\Delta}_k = \delta_k D \eta_x$ where $\delta_k$ is obtained as the maximum of the reduced functional

$$\widetilde{\mathcal{F}}[\delta_k] := \frac{\sum_{k,k'} \chi_{k-k'} \delta_k^* \delta_{k'}}{\sum_k |\delta_k|^2}, \qquad (13)$$

or equivalently as the largest eigenvector of $\chi_{k-k'}$ viewed as a matrix in $k$ and $k'$. As $\chi_{k-k'} > 0$ (due to stability), the Perron-Frobenius theorem then immediately implies $\delta_k > 0$, in line with out assumption above and as stated in the main text.

### Electron–phonon coupling
To present more details on the electron–phonon coupling, the associated displacement operators in Eq. (8) can be expressed in terms of canonical bosons, $b_{g,\alpha,\mu,q}$,

$$(u_{g,\mu}(r))_j = \sum_q \frac{b_{g,j,\mu,q} e^{iq \cdot r} + \mathrm{H.c.}}{\sqrt{2NM\omega_g(q)}}, \qquad (14)$$

where $j$ refers to the two components for the $E_2$ phonon (is idle for $A_1, B_1$), $M$ is the carbon mass, and $\omega_g(q)$ is the phonon dispersion, characterizing the phononic part of the Hamiltonian, $\mathcal{H}_P = \sum_q \omega_g(q) b_{g,j,\mu,q}^\dagger b_{g,j,\mu,q}$.

As for $(v_\mu)_\ell$ in Eq. (8), $\ell = 1, 2$ refers to the physical graphene layer in the case of TBG. One can, in principle, choose any orthonormal basis; we will find it convenient to use the layer-exchange even and odd states, $v_\pm = (1, \pm 1)^T / \sqrt{2}$. For TTG, the situation is more involved (see Supplementary Appendix D2), but our arguments about which phonons are attractive in which pairing channels will hold for both systems.

We project $\mathcal{H}_{EP}$ in Eq. (8) onto the two flat bands ($\alpha = \pm$) in each valley $\eta$ of the spin-polarized continuum-model, leading to a coupling term similar to Eq. (4) with momentum-dependent coupling matrices, $\lambda^j \to \lambda^{g,j,\mu}_{k,k'}$. Investigating the matrix elements $\lambda^{g,j,\mu}_{k,k'}$, we notice that they almost vanish for the layer-odd intervalley ($A_1, B_1$) phonons, which can be understood as a consequence of chiral and particle-hole symmetry (see Supplementary Appendix D3). The situation is the reverse for the intra-valley ($E_2$) phonons, where the layer-even matrix elements are numerically small and the layer-odd matrix elements dominate. We therefore focus on layer-even (odd) intervalley (intra-valley) phonon couplings.

Neglecting the momentum dependence in the phonon frequencies and retardation effects, the resulting electron-electron interaction in the inter-valley Cooper channel obtained by integrating out the phonons is given by Eq. (9). Here, $V_g = g_g^2 / (2N\omega_g^2) > 0$ and $V_{A_1} = V_{B_1} \simeq 1.33 V_{E_2}$ results from $g_{A_1} = g_{B_1} \simeq g_{E_2}$ and the phonon frequencies estimated in ref. 67. Importantly, this only holds for parallel spins in the two valleys. For anti-parallel spins, the projection of the coupling matrices to the flat bands vanishes for the intervalley phonon modes $A_1$ and $B_1$ such that $V_{A_1} = V_{B_1} = 0$.

## Data availability

The data generated in this study are available in the Zenodo database under the accession code https://zenodo.org/record/8381555and in the figshare repository https://doi.org/10.6084/m9.figshare.23897019.

## Code availability

The codes used to generate the plots are available from the corresponding author on request.

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

## Acknowledgements
M.S.S. acknowledges funding from the European Union (ERC-2021-STG, Project 101040651—SuperCorr). Views and opinions expressed are however those of the authors only and do not necessarily reflect those of the European Union or the European Research Council Executive Agency. Neither the European Union nor the granting authority can be held responsible for them. M.C. and S.S. acknowledge funding by U.S. National Science Foundation grant No. DMR-2002850. M.S.S. Thanks B. Putzer for the discussions. M.C. thanks P. Ledwith, J. Dong, and D. Parker for their helpful discussions.

## Author contributions
M.C., S.S., and M.S.S. contributed to the research. M.C. and M.S.S. performed the numerical computations and wrote the paper.

## Funding

## Competing interests
The authors declare no competing interests.
