## [Peer Review File · Nature Communications]

Nodal band-off-diagonal superconductivity in twisted graphene superlatticesReviewers' Comments:

Reviewer #1:

Remarks to the Author:

Christos et al. address the possible superconducting states in twisted bilayer and trilayer graphene. They use symmetry arguments and many-body theory applied to models for the flat bands, including pairing interactions via the exchange of phonons and of fluctuations related to plausible order parameters. There are several remarkable results: Due to the combination of spin polarization and fermionic antisymmetry, pairing states that are even under the twofold rotation about the normal axis are odd and thus completely off-diagonal in the band index, i.e., they realize pure interband pairing. Conversely, pairing states that are odd under this rotation are even in the band index. Various pairing states, in particular the former ones, have guaranteed nodes that are not expected based on their irreducible representation (irrep). Remarkably, the fully symmetric A_1 state realizes pure interband pairing and has such nodes. Apart from point nodes, relevant pairing states have line nodes, which are the analog of Bogoliubov Fermi surfaces in the two-dimensional systems under consideration. The presence and character of nodes for some pairing states depends on the magnitude of the pairing amplitude. Numerical evaluation shows that an exotic state with pure interband pairing and line nodes can be energetically favored, although this cannot be a weak-coupling instability.

Twisted graphene superlattices are an important field of research, not only because of the complex phase diagrams observed for these systems, but also because they realize a new paradigm of how to design strongly correlated electron systems. The nature of the observed superconducting states is not well understood. The present manuscript takes a large step in understanding these states. The results are rather unexpected but overall well justified and will certainly motivate further theoretical and experimental research. In my view, all this makes the manuscript suitable for Nature Communications, in principle.

I have a number of comments, though. Most importantly, I think that for the broad readership of Nature Communications it does not become sufficiently clear how truly exotic the superconducting states are. The abstract mentions line nodes but it is very easy to miss the essential point that these are Bogoliubov Fermi surfaces in two dimensions. Experimental evidence for superconductors with Bogoliubov Fermi surfaces is scarce and I do not know of a compelling case in the absence of an applied magnetic field. Similarly, it is remarkable that pure interband pairing, which does not have a weak-coupling instability (as the authors say rather late), can be favored over less exotic states that do have one. I thus recommend that the authors revise the abstract, the introduction, and the conclusions in a way that better explains the novelty of the results to a broad audience.

Additional points:

- (1) I find the designation of the E_2 state as "chiral d-wave" misleading. The lowest-order polynomial basis functions are (k_x, k_y) . Hence, I see this rather as a band-off-diagonal chiral p-wave state.
- (2) The abbreviations for the (non-superconducting) order parameters are introduced at various places and "K-IVC" and "SLP⁻" are not defined at all. I recommend to insert a table defining all of them. The sentence "We also show that fluctuations of both T-IVC..." starting in line 84 is confusing for readers who are unfamiliar with these order parameters. Also, the position of the +/- sign in SLP⁺ and SLP⁻ is inconsistent between the text, Table II, and Fig. 4.
- (3) Table I: The caption should specify that the hat refers to 2x2 matrices. A minus sign is missing in the header of the second column.
- (4) The expression in line 144 is sloppy; σ_z should be $(\sigma_z)_\alpha$, which equals α .
- (5) In line 240-241, the authors state that vorticity -1 is inconsistent with threefold rotation symmetry. But surely the $X - iY$ state has vorticity -1?

(6) I do not understand the statement in line 308. D commutes with λ^x but not with λ^y .

(7) The discussion at the end of Sec. II.D seems vague and confusing to me. The authors seem to say that their theory does not reproduce the experiments. This requires forthright statements and a clear discussion of how their work is related to real materials.

(8) The symbol ρ_j , $j = x,y,z$, first used in line 405, is not defined in the text, only in the supplement.

(9) The horizontal axis of Fig. 3(c) lacks a label. In the caption, the authors should check whether the final "(b)" should be "(a)".

(10) The caption of Table II is not understandable by itself. Also, it is confusing that the entry "B₂, B₁" in the last row and column seems to contradict Fig. 4.

(11) The discussion starting in line 499 does not really make clear how each $\bar{\lambda}^j$ is chosen and how it is related to the λ^j listed in the same row in Table II.

(12) The last sentence of Sec. II.F is correct as written but I do not understand whether the situation described here is generic or requires fine tuning.

Carsten Timm

Reviewer #2:

Remarks to the Author:

In the manuscript "*Nodal band-off-diagonal superconductivity in twisted graphene superlattices*"

the authors study the features of the superconducting (SC) order parameter in the electron-doped, spin-polarized twisted bilayer (TBG) and trilayer graphene (TTG).

The authors show that the spin polarization allows for a purely off-diagonal pairing in the band basis,

which occurs when the C_{2z} symmetry is preserved.

Notably, such off-diagonal structure of the SC order parameter is consistent with both phonon-mediated pairing and pairing induced by the fluctuations of a time-reversal-symmetric inter-valley coherent order.

The SC gap can have nodal lines or the Fermi surface can be completely gapped out, depending on the parameters,

allowing for a V- or a U-shaped density of states.

The whole analysis is supported by mean-field (Hartree-Fock) calculations performed on top of the flat-bands of the TBG and the TTG.

The results are able to reproduce a variety of spectral features reported experimentally in the SC TBG and TTG, leading the authors to identify the band-off-diagonal pairing as a key feature of these kind of systems.

I find the work original and the results clearly and well presented. Even though I have no peculiar remarks, I think that the manuscript is too technical for a journal like *Nature Communications*, which is devoted to a broad audience. I then recommend the authors to submit the manuscript to a more specific journal.

I have a minor comment:

The authors mention that the Fig. 2(a) represents the band structure for filling $\nu=2$. However, they say that the calculations have been performed for filling $\nu=2.5$ ν that, to the naked eye, seems to be the value of the filling in the Fig. 2(a). Is it just a misprint or is the Fig. 2(a) actually referring to $\nu=2$?

Reviewer #3:

Remarks to the Author:

In the submitted manuscript entitled, "Nodal band-off-diagonal superconductivity in twisted graphene superlattice," the authors theoretically study the nature of superconducting pairing in twisted bilayer graphene. Specifically, the authors propose that due to symmetry constraints of flavor polarization, the pairing channels for superconductivity are necessarily band-off-diagonal. This results in an order parameter, which can be a chiral d-wave in line with recent experimental results, that can, depending on interactions, be either gapped or nodal. The authors propose this to explain recent STM measurements from the Yazdani and Nadj-Perge groups. The authors also demonstrate that this off-diagonal mechanism is possible for both phonon-mediated pairing and fluctuation-mediated pairing of the IVC order parameter.

A crucial assumption of the authors' analysis, I also believe the most contentious part, is that the normal state from which SC emerges is spin-polarized. To what extent that is the case experimentally remains to be verified. Provided that this assumption holds, then the authors' analysis is rigorous. The paper is well-written and highly informative, and I do not have any technical criticisms. I think that the paper will be of interest to the moiré community, particularly the theorists. As such, I believe it is in line with Nature Communication standards. I am happy to recommend its publication. I invite the authors to consider adding a few clarifying comments to the text, which I think can strengthen the manuscript's impact. My comments are:

- Majority (if not all) experimental samples are strained. To what extent can the authors apply in such a context, where I imagined the symmetries of the order parameters are less restrictive? Can the authors comment on it in the manuscript? This is important as the $\nu=-2$ in strained samples of Yazdani and Nadj-Perge is most likely a spin-unpolarized IKS.

- The authors analyze two mechanisms for pairing, which I agree are the most likely types. Is there an experimental prediction that the authors can make which can be used to disentangle the two candidates? For example, dip-hump features in the tunneling conductance (See Nadj-Perge paper, for instance), Position dependence inside the moiré cell, etc. More generally, the reason for my question is that although the author's work is highly comprehensive and provides a plausible scenario to explain existing experiments, it is still a postdiction. Is there an experimentally testable prediction that the authors can make?

- Can the authors comment in the text (or calculate) on the expected form of the Andreev tunneling, which their mechanism would predict? Specifically, I imagine that the Andreev signal would increase in the U-shaped region compared to the V-shape region as there are no electron states which would increase reflection. Moreover, this indeed would be a fascinating consequence of the authors' work. Would the order parameter value (e.g., the gap at the antinodal point of the d-wave order, for example) as measured in the Andreev tunneling be comparable to the gap as measured in the STM or different? The reason for my question, as I am sure the authors are aware, is that there is a discrepancy in these two gaps as measured from STM and QPC by the Yazdani and Nadj-Perge groups, which remains unexplained.

- For order parameter fluctuations and phonon-mediated pairing, the authors take a particular kernel form, specifically $\chi(q)$. My question is how do the values of the parameters (e.g., V in line 511, for example) compare to the expected values for phonon-mediated pairing (if, for example, it was coming from the optical graphene phonons as hinted in Ref. [61]. Using realistic parameter values what mean-field T_C would the authors get. How does it compare to the T_* (pseudogap temperature scale) seen experimentally?

- Can the authors comment and argue why and to what extent a weak-coupling calculation (i.e., just a gap equation that neglects self-energy corrections, etc.) applies for modeling SC pairing in MATBG? I understand that this is a difficult question theoretically, but high importance is placed on the strong-coupling nature of moiré superconductivity, and thus, a qualification of what claims authors make that they expect to persist would be valuable.

- I like the argument the authors make as a possible explanation for the U to V evolution, but the tunneling spectra in Fig. 2E have a lot of features (oscillations). Could the authors explain what features of the DOS are coming from the underlying bandstructure and what is a property of the superconductor? Could the authors comment on the lack of visible coherence peaks (unlike the experimental measurements) and whether there are parameters in their simulation that control the sharpness of the coherence peaks? Perhaps it may be helpful to plot the tunneling spectra of Fig. 2E normalized by the normal state conductance to "cancel out" bandstructure features?

- Minor: just a notation convention: throughout the whole paper in bandstructure plots, authors use a notation of high symmetry points (e.g., K, K', Gamma, etc.). Could the authors qualify and distinguish high-symmetry points of moiré BZ from the graphene BZ?

- Minor: Fig. 1B. BEC/BI is not defined. I assume Bose-Einstein Condensate/Band Insulator, but it may be helpful to explain.

- Minor: line 388. Do the authors mean the Dynes formula?

- Minor: Eq. S12. I don't expect this to change any of the author's results qualitatively, but the form of the screened potential used is for double-gated devices, while the STM works are single-gated. The form of the potential would then be (obtained from solving Poisson's equation for vacuum - material at $z = 0$ - dielectric of thickness d and dielectric constant ϵ - metal gate), is $\frac{e^2}{2\epsilon_0 q} \frac{2}{1 + \epsilon_r \coth(qd)}$ with $\epsilon_r = \epsilon / \epsilon_0$. This potential is less screened at a small q than the one the authors use.

Resubmission of NCOMMS-23-16329:

**“Nodal band-off-diagonal superconductivity in twisted graphene
superlattices”**

by Maine Christos, Subir Sachdev, and Mathias S. Scheurer

(Dated: August 7, 2023)

Response to Reviewer # 1:

Reviewer: *Christos et al. address the possible superconducting states in twisted bilayer and trilayer graphene. They use symmetry arguments and many-body theory applied to models for the flat bands, including pairing interactions via the exchange of phonons and of fluctuations related to plausible order parameters. There are several remarkable results: Due to the combination of spin polarization and fermionic antisymmetry, pairing states that are even under the twofold rotation about the normal axis are odd and thus completely off-diagonal in the band index, i.e., they realize pure interband pairing. Conversely, pairing states that are odd under this rotation are even in the band index. Various pairing states, in particular the former ones, have guaranteed nodes that are not expected based on their irreducible representation (irrep). Remarkably, the fully symmetric A_1 state realizes pure interband pairing and has such nodes. Apart from point nodes, relevant pairing states have line nodes, which are the analog of Bogoliubov Fermi surfaces in the two-dimensional systems under consideration. The presence and character of nodes for some pairing states depends on the magnitude of the pairing amplitude. Numerical evaluation shows that an exotic state with pure interband pairing and line nodes can be energetically favored, although this cannot be a weak-coupling instability.*

Twisted graphene superlattices are an important field of research, not only because of the complex phase diagrams observed for these systems, but also because they realize a new paradigm of how to design strongly correlated electron systems. The nature of the observed superconducting states is not well understood. The present manuscript takes a large step in understanding these states. The results are rather unexpected but overall well justified and will certainly motivate further theoretical and experimental research. In my view, all this makes the manuscript suitable for Nature Communications, in principle.

Response: We thank Reviewer# 1 for his careful reading of our manuscript and for his positive recommendation.

Reviewer: *I have a number of comments, though. Most importantly, I think that for the broad readership of Nature Communications it does not become sufficiently clear how truly exotic the superconducting states are. The abstract mentions line nodes but it is very easy to miss the essential point that these are Bogoliubov Fermi surfaces in two dimensions. Experimental evidence for superconductors with Bogoliubov Fermi surfaces is scarce and I do not know of a compelling case*

in the absence of an applied magnetic field. Similarly, it is remarkable that pure interband pairing, which does not have a weak-coupling instability (as the authors say rather late), can be favored over less exotic states that do have one. I thus recommend that the authors revise the abstract, the introduction, and the conclusions in a way that better explains the novelty of the results to a broad audience.

Response: After rereading the manuscript, we agree with the referee that our previous version did not sufficiently emphasize the surprising consequences of our A_2 pairing and the associated Bogoliubov Fermi surfaces it exhibits. We thank him for this helpful feedback and we have revised our abstract, introduction, and conclusion to emphasize the unconventional aspects of our A_2 pairing phenomenology and energetics appropriately.

Reviewer: *Additional points:*

(1) I find the designation of the E_2 state as “chiral d-wave” misleading. The lowest-order polynomial basis functions are (k_x, k_y) . Hence, I see this rather as a band-off-diagonal chiral p-wave state.

Response: We thank the reviewer for his suggestion to change “chiral d-wave” to “chiral p-wave”. We agree this is a more natural label and we have changed it where it appears in the text.

(2) The abbreviations for the (non-superconducting) order parameters are introduced at various places and “K-IVC” and “ SLP^- ” are not defined at all. I recommend to insert a table defining all of them. The sentence “We also show that fluctuations of both T-IVC...” starting in line 84 is confusing for readers who are unfamiliar with these order parameters. Also, the position of the +/- sign in SLP^+ and SLP^- is inconsistent between the text, Table II, and Fig. 4.

Response: At the reviewer’s suggestion, we have added a brief explanation of the SLP^+ , SLP^- , K-IVC, and N-IVC phases at the beginning of Section II F. We also explicitly reference Table II, where the respective order parameters are explicitly stated, in the sentence in the introduction quoted by the referee. Furthermore, we have made the notation for SLP^\pm consistent throughout the main text and SI. Thanks for noticing this.

Reviewer: (3) *Table I: The caption should specify that the hat refers to 2x2 matrices. A minus sign is missing in the header of the second column.*

Response: We thank the reviewer for pointing out the typo, which we have corrected in the revised manuscript. We also followed his helpful suggestion and explicitly mention that the basis functions with hats are 2×2 matrices.

Reviewer: (4) *The expression in line 144 is sloppy; σ_z should be $(\sigma_z)_{\alpha\alpha}$, which equals α .*

Response: We thank the reviewer for pointing this out, and we have corrected this typo in our revised manuscript.

Reviewer: (5) *In line 240-241, the authors state that vorticity -1 is inconsistent with threefold rotation symmetry. But surely the $X - iY$ state has vorticity -1?*

Response: We thank the referee for pointing out that the comment in parentheses was misleading. What we wanted to say is that, although the basis functions $X_{\mathbf{k}}$ and $Y_{\mathbf{k}}$ are only constrained by a discrete rotational symmetry (rather than a continuous rotational symmetry), the vorticity v of $X_{\mathbf{k}} + iY_{\mathbf{k}}$ can only be $v \in 3\mathbb{Z} + 1$ (which does not include $v = -1$). This, however, is not important at all for our analysis and, in fact, even confusing in the current context of spontaneous symmetry breaking. We have removed that comment in the manuscript.

Reviewer: (6) *I do not understand the statement in line 308. D commutes with λ^x but not with λ^y .*

Response: While this is true, Eq. 7 involves the commutator of $D\eta_x$ (and not D) with $\lambda^{x,y}$. So, in the context of the statement in (previous) line 308, we have $D = \sigma_y\eta_x$, and therefore $D\eta_x = \sigma_y$, which commutes with both $\lambda^x = \eta_x$ and $\lambda^y = \eta_y$ and thus Eq. 7 is satisfied for A_2 pairing and T-IVC fluctuations.

Reviewer: (7) *The discussion at the end of Sec. II.D seems vague and confusing to me. The authors seem to say that their theory does not reproduce the experiments. This requires forthright statements and a clear discussion of how their work is related to real materials.*

Response: Upon reconsidering it, we agree with the referee that the previous discussion at the end of Sec. II.D was vague and confusing. We have revised it accordingly to make it more concise and clear in the revised version of the manuscript: in short, our theory provides a natural explanation for the two regimes seen in experiment characterized by a U-shaped and a V-shaped DOS, and even their doping dependence is natural. Within our theory, there can also be a regime with a peak in the DOS at low energies which is not reported in current experiment. However, this regime is very small in parameter space (just before superconductivity is energetically disfavored) and strongly depends on the normal state. So it might not be there in the real system (due to additional perturbations, like disorder, or corrections beyond our modeling) or might be very difficult to find in experiment.

We also noticed that using V instead of “interaction strength” can sometimes be confused with the “V” in “V-shaped”. We changed the wording to make it clear.

Reviewer: (8) *The symbol ρ_j , $j = x,y,z$, first used in line 405, is not defined in the text, only in the supplement.*

Response: At Reviewer #1’s suggestion, we have clearly defined ρ_j where it is introduced in the main text.

Reviewer: (9) *The horizontal axis of Fig. 3(c) lacks a label. In the caption, the authors should check whether the final “(b)” should be “(a)”.*

Response: At Reviewer #1’s suggestion, we have made the above mentioned corrections to Fig. 3 and its caption. Thanks for catching these typos.

Reviewer: (10) *The caption of Table II is not understandable by itself. Also, it is confusing that the entry “ B_2, B_1 ” in the last row and column seems to contradict Fig. 4.*

Response: Let us first resolve what seems to be a contradiction: the last column in Table II refers to the superconducting states that dominate for interactions with fermion-boson coupling listed in the second column of Table II (labeled λ^j). This follows immediately from Eq. (7) and, when using

these interactions in our numerics, we also find consistent results. However, to provide numerics that complements these conclusions in a more non-trivial, we have used the couplings listed in the third column ($\bar{\lambda}^j$) for the results in Fig. 4. While, in each row, λ^j and $\bar{\lambda}^j$ have the same symmetries (and thus correspond to the same phase), they are *not* identical: For instance, projecting $\bar{\lambda}^j$ to the flat-band basis, as we do in our numerics, produces momentum-dependent coupling vertices in the flat-band basis, while λ^j are by design momentum independent in that basis. It turns out that in most cases, the leading superconductor is still the same, with the SLP– (last row) being the only exception in Table II. We note that this can even be understood analytically by applying Eq. (7) in the sublattice basis, as we explain in the SI (see Table S1).

While we do mention this in the main text, we agree with the referee that this is not clear when only looking at the caption of Table II (and Fig. 4), which we have extended accordingly in the revised manuscript.

Reviewer: (11) *The discussion starting in line 499 does not really make clear how each $\bar{\lambda}^j$ is chosen and how it is related to the λ^j listed in the same row in Table II.*

Response: Each of those phases is characterized by certain symmetries. We mention these more explicitly in the revised version of Sec. II.F but refer, e.g., to Table II in PRX **12**, 021018 (2022) for a detailed list of the symmetries (which we also cite now more explicitly). For $\bar{\lambda}^j$, we just write down the momentum-independent combination of Pauli matrices in the microscopic sublattice basis with the same symmetries as the corresponding λ^j , expressed in the band basis. In this sense, λ^j and $\bar{\lambda}^j$ describe coupling of an order parameter of the same phase but their precise microscopic form is different: while λ^j is (by design) momentum-independent in the band basis, $\bar{\lambda}^j$ is not. We agree with the referee that this was not clear in the previous version of the manuscript and have revised the explanation accordingly.

Reviewer: (12) *The last sentence of Sec. II.F is correct as written but I do not understand whether the situation described here is generic or requires fine tuning.*

Response: We reworded the last sentence in Sec. II.F to clarify that the nematic E_1 pairing being nodal only requires that the band off diagonal part of the pairing is sufficiently small relative to the bandwidth, but no fine tuning once this condition is met.

Response to Reviewer # 2:

Reviewer: *In the manuscript “Nodal band-off-diagonal superconductivity in twisted graphene superlattices” the authors study the features of the superconducting (SC) order parameter in the electron-doped, spin-polarized twisted bilayer (TBG) and trilayer graphene (TTG). The authors show that the spin polarization allows for a purely off-diagonal pairing in the band basis, which occurs when the C_{2z} symmetry is preserved. Notably, such off-diagonal structure of the SC order parameter is consistent with both phonon-mediated pairing and pairing induced by the fluctuations of a time-reversal-symmetric inter-valley coherent order. The SC gap can have nodal lines or the Fermi surface can be completely gapped out, depending on the parameters, allowing for a V- or a U-shaped density of states. The whole analysis is supported by mean-field (Hartree-Fock) calculations performed on top of the flat-bands of the TBG and the TTG. The results are able to reproduce a variety of spectral features reported experimentally in the SC TBG and TTG, leading the authors to identify the band-off-diagonal pairing as a key feature of these kind of systems.*

I find the work original and the results clearly and well presented. Even though I have no peculiar remarks, I think that the manuscript is too technical for a journal like Nature Communications, which is devoted to a broad audience. I then recommend the authors to submit the manuscript to a more specific journal.

Response: We thank reviewer #2 for carefully reading our manuscript and for their summary of our work, as well as their positive comments, stating that our work is “original” and “the results clearly and well presented”; we are also happy to read that they share our view that our “results are able to reproduce a variety of spectral features reported experimentally in the SC TBG and TTG”.

With respect to their point that our paper may be too technical for publication in Nature Communications; we have added additional statements to the abstract and introduction which emphasize the highly unusual phenomenology of our proposed A_2 pairing and its associated Bogoliubov Fermi surfaces and also the equally surprising fact that such a pairing is not only possible but energetically favorable due to the system’s unique combination of band topology and set of nearly dimensionless bands. To understand these conclusions one does not have to go through the entire algebra of our work nor understand the technicalities. We have further removed the most technical part of the previous version of the manuscript—the definition of the electron-phonon coupling—to the Methods section. We have also added two subfigures to our Figure 1 which makes the nodal

nature of our interband pairing more intuitive for readers who are unfamiliar with Bogoliubov Fermi surfaces. We are confident that these modifications make our manuscript even more accessible and clearly appropriate for the broad graphene community and beyond.

Reviewer: *I have a minor comment: The authors mention that the Fig. 2(a) represents the band structure for filling $\nu = 2$. However, they say that the calculations have been performed for filling $\nu = 2.5$ that, to the naked eye, seems to be the value of the filling in the Fig. 2(a). Is it just a misprint or is the Fig. 2(a) actually referring to $\nu = 2$?*

Response: For the plot of the band structure in Fig. 2(a), we have set the chemical potential such that there is filling $\nu = 2.5$ per moiré unit cell. However, the interaction renormalization was computed at $\nu = 2$, and we expect that varying the chemical potential will not qualitatively change our results since the most significant determining factor in existence of an A_2 pairing solution is the particle-hole symmetric band splitting, and not the particle-hole symmetry-breaking terms like the chemical potential and Hartree band renormalization. We have clarified this in Sec. II.D in the revised version of the manuscript.

Response to Reviewer # 3:

Reviewer: *In the submitted manuscript entitled, “Nodal band-off-diagonal superconductivity in twisted graphene superlattice,” the authors theoretically study the nature of superconducting pairing in twisted bilayer graphene. Specifically, the authors propose that due to symmetry constraints of flavor polarization, the pairing channels for superconductivity are necessarily band-off-diagonal. This results in an order parameter, which can be a chiral d-wave in line with recent experimental results, that can, depending on interactions, be either gapped or nodal. The authors propose this to explain recent STM measurements from the Yazdani and Nadj-Perge groups. The authors also demonstrate that this off-diagonal mechanism is possible for both phonon-mediated pairing and fluctuation-mediated pairing of the IVC order paper.*

A crucial assumption of the authors’ analysis, I also believe the most contentious part, is that the normal state from which SC emerges is spin-polarized. To what extent that is the case experimentally remains to be verified. Provided that this assumption holds, then the authors’ analysis is rigorous. The paper is well-written and highly informative, and I do not have any technical criticisms. I think that the paper will be of interest to the moiré community, particularly the theorists. As such, I believe it is in line with Nature Communication standards. I am happy to recommend its publication. I invite the authors to consider adding a few clarifying comments to the text, which I think can strengthen the manuscript’s impact.

Response: We thank the reviewer for their careful reading and the positive evaluation of our manuscript and are happy to hear that they recommend publication in Nature Communications. We also thank the referee for their constructive and useful comments, which we address in the following response and in the revised manuscript.

Reviewer: *My comments are: - Majority (if not all) experimental samples are strained. To what extent can the authors apply in such a context, where I imagined the symmetries of the order parameters are less restrictive? Can the authors comment on it in the manuscript? This is important as the $\nu = -2$ in strained samples of Yazdani and Nadj-Perge is most likely a spin-unpolarized IKS.*

Response: Our central conclusions are not affected by the additional presence of strain, which breaks the C_{3z} symmetry and, hence, reduces the point group from C_6 to C_2 : as the central C_{2z} will

still be present, the fact that certain pairing states are protected to be entirely band-off-diagonal is not affected. More specifically, C_2 has two irreducible representations (IRs), A (even under C_{2z}) and B (odd), and all states above (below) the horizontal line in Table I will transform under A (B). This means that if, say, the A_2 superconductor is favored without strain, breaking C_{3z} by strain (or other sources for that matter) will lead to the admixture of A_1 and E_2 pairing (in a way that does not break time-reversal symmetry, since A is a one-dimensional IR). The resulting superconductor will still have the same unconventional spectral properties and could explain the transition from U- to V-shaped tunnel DOS. Since this is a relevant point, we have added a comment in Sec. II.A of the revised manuscript. We finally note that we agree with the referee that the normal state properties are indeed a crucial ingredient. While, in our view (see also Phys. Rev. B **106**, 104506), some form of spin order is most natural, we have extended our analysis to superconductivity co-existing with T-IVC order (see Appendix E in the revised version of the manuscript), where again pairing channels exist that are entirely band-off-diagonal, can dominate energetically, and which exhibit similar physics.

Reviewer: - *The authors analyze two mechanisms for pairing, which I agree are the most likely types. Is there an experimental prediction that the authors can make which can be used to disentangle the two candidates? For example, dip-hump features in the tunneling conductance (See Nadj-Perge paper, for instance), Position dependence inside the moiré cell, etc. More generally, the reason for my question is that although the author's work is highly comprehensive and provides a plausible scenario to explain existing experiments, it is still a postdiction. Is there an experimentally testable prediction that the authors can make?*

Response: The reviewer is asking whether there may be some experimental signature which can distinguish between fluctuation- and phonon-mediated pairing. As we point out in section E, the A_1 and B_1 phonons which uniquely pick out our inter-band A_2 pairing are intervalley, and therefore can only serve as the pairing glue in the case where the bands are spin polarized. Therefore, if the flavor polarization is identified to be spin-valley locked and our proposed A_2 state is realized, the dominant pairing mechanism is most likely coming from fluctuations of (a spinful version of) T-IVC order. We also point out that because T-IVC fluctuations and A_1 phonons behave the same under the symmetries of TBG, in the case of spin polarized bands, they may act together to stabilize the A_2 pairing, and in the case of spin polarization may not be distinguishable.

We also point out the following possible scenario; if the normal state is spin-valley locked, we

expect A_1 and B_1 phonons will not play a role, as mentioned above. However, in the presence of a magnetic field, our expectation is the spin orientation of the zero field spin valley locked state will cant in the direction of the magnetic field as field increases, and it is therefore possible that in high fields, A_1 and B_1 phonons may reemerge as a possible pairing glue, an observation that is not inconsistent with the presence of a re-entrant superconducting phase [Nature **595**, 526–531 (2021)] at high fields in twisted trilayer graphene (in twisted bilayer graphene this would likely be overcompensated by the in-plane orbital coupling). Identification of the pairing symmetry in either the low or high field phases would serve as indirect evidence for or against this hypothesis. We have noted these observations at the end of our section on phonon mediated pairing and thank the referee for the question.

Reviewer: - *Can the authors comment in the text (or calculate) on the expected form of the Andreev tunneling, which their mechanism would predict? Specifically, I imagine that the Andreev signal would increase in the U-shaped region compared to the V-shape region as there are no electron states which would increase reflection. Moreover, this indeed would be a fascinating consequence of the authors' work. Would the order parameter value (e.g., the gap at the antinodal point of the d-wave order, for example) as measured in the Andreev tunneling be comparable to the gap as measured in the STM or different? The reason for my question, as I am sure the authors are aware, is that there is a discrepancy in these two gaps as measured from STM and QPC by the Yazdani and Nadj-Perge groups, which remains unexplained.*

Response: The reviewer has asked us to comment on what experimentally measured Andreev reflection signal should be expected for our proposed purely band off diagonal pairing. While we would agree with their assertion that a U-shaped region would have a larger Andreev signal, previous computations done of an STM tip in the strong coupling regime rely on the assumption of a single band BdG dispersion which is qualitatively different from the multiband case. Computing the Andreev reflection amplitude in the multi-band case requires a more involved calculation beyond the scope of our current work and therefore we cannot reliably predict something to be compared to Yazdani's and Nadj-Perge's experimental measurements. While we thank the referee for the inspiring question, which we want to look into in a follow-up project, we do not want to speculate about it in the manuscript to avoid making unjustified (or potentially incorrect) statements. We only mention it as an outlook at the end.

Referee: - *For order parameter fluctuations and phonon-mediated pairing, the authors take a particular kernel form, specifically $\chi(\mathbf{q})$. My question is how do the values of the parameters (e.g., V in line 511, for example) compare to the expected values for phonon-mediated pairing (if, for example, it was coming from the optical graphene phonons as hinted in Ref. [61]). Using realistic parameter values what mean-field T_C would the authors get. How does it compare to the T_* (pseudogap temperature scale) seen experimentally?*

Response: For T-IVC fluctuations as shown in Fig. 2, we assume the matrix elements of the interactions take on the simple, momentum independent form of the T-IVC order parameter in the flat bands; in this case there is no natural scale to choose for interactions since both the strength of fluctuations and their momentum dependence are unknown, so we choose V to be large enough such that the $T = 0$ A_2 state is stabilized. We note that while the V we choose for T-IVC fluctuations is larger than the rough scale of phonon interactions, it is still less than the scale of Coulomb interactions. The parameters of $\chi(\mathbf{q})$ also have no known estimates experimentally and theoretically. We choose a small lifetime for our Lorentzian to make it theoretically harder for the A_2 state to win energetically; all else being equal we would expect a more sharply peaked potential would be favorable to any sub-leading states which have sign changes and if anything should make them more competitive with our A_2 state. For optical phonons, we use a coupling of $250 \text{ meV}\cdot\text{nm}^2$ to be compared with the estimated theoretical value of $60 - 70 \text{ meV}\cdot\text{nm}^2$. However, for phonons this value is chosen to demonstrate what the A_2 band structure looks like just as the pairing is becoming fully gapped and we have checked that an $80 \text{ meV}\cdot\text{nm}^2$ coupling is sufficient to stabilize the A_2 pairing if the normal state is taken to be the continuum model. Our estimated T_c depends linearly on the band splitting; for flat bands and phonon interactions, our T_c is roughly on the order of 5 K. However for T-IVC fluctuations where we employ a Coulomb renormalized band structure which has a band splitting much larger than the continuum model, the T_c is $\simeq 10$ K. Thus the expected minimum allowed T_c for A_2 pairing is likely somewhere between these two temperatures, with the precise value depending strongly on the normal state band structure. Our computed T_c given this rough range would therefore likely be on the order of the pseudo-gap T_* which persists at least up to 6.7 K.

Referee: - *Can the authors comment and argue why and to what extent a weak-coupling calcula-*

tion (i.e., just a gap equation that neglects self-energy corrections, etc.) applies for modeling SC pairing in MATBG? I understand that this is a difficult question theoretically, but high importance is placed on the strong-coupling nature of moiré superconductivity, and thus, a qualification of what claims authors make that they expect to persist would be valuable.

Response: The reviewer raises the question of what effects we may be overlooking by restricting our numerics to solving a mean-field gap equation. We first point out in the trilayer system at doping closer to $\nu = 3$, the superconductor displays properties in its transition temperature and coherence length consistent with BCS theory, and at least in this regime a mean-field approach may be justified. Furthermore, albeit not exact, mean-field often gives very good qualitative insights which phases are favored, in particular, for superconductivity. However, we acknowledge there are several effects our approach does not account for; as we note in our conclusion, a more exact approach, which allows for frequency dependent pairing, like Eliashberg theory, seems like an important next step. In fact, we are already working on a followup project using Eliashberg theory. At the time of writing this response (but note this is ongoing research), it seems that the interband phase we propose in the current manuscript seems to be also favored within Eliashberg theory in a wide range of parameters. We are also planning future work to answer the question of whether any exact superconducting ground states may be constructed for either phonon or T-IVC interactions in the limit of zero dispersion.

Referee: - *I like the argument the authors make as a possible explanation for the U to V evolution, but the tunneling spectra in Fig. 2E have a lot of features (oscillations). Could the authors explain what features of the DOS are coming from the underlying bandstructure and what is a property of the superconductor? Could the authors comment on the lack of visible coherence peaks (unlike the experimental measurements) and whether there are parameters in their simulation that control the sharpness of the coherence peaks? Perhaps it may be helpful to plot the tunneling spectra of Fig. 2E normalized by the normal state conductance to “cancel out” bandstructure features?*

Response: The oscillations referenced by Reviewer # 3 are primarily from the normal-state band structure; there are two distinct peaks on either the positive or negative energy side of the DOS corresponding to the two renormalized flat bands. The separation of the peaks depends on a combination of the initial normal state band splitting and additional renormalization effects from the

superconductor. While we cannot say with certainty why there is a lack of coherence peaks, we suspect it may likely be due to the lack of level repulsion in the superconducting state between the two lower bands; in a fully gapped intra-band superconductor the bands nearest to the Fermi level would likely be more flattened and lead to a sharper VHS signature in the density of states. We also cannot rule out insufficient k-space resolution in our microscopic calculations, as this may also “wash out” any coherence peaks which may be observable with a finer momentum grid.

Response: - *Minor: just a notation convention: throughout the whole paper in bandstructure plots, authors use a notation of high symmetry points (e.g., K , K' , Γ , etc.). Could the authors qualify and distinguish high-symmetry points of moiré BZ from the graphene BZ?*

Response: The high symmetry points listed above are indeed referring to high symmetry points of the moiré scale Brillouin zone. We have clarified this in the caption for Fig. 2.

Response: - *Minor: Fig. 1B. BEC/BI is not defined. I assume Bose-Einstein Condensate/Band Insulator, but it may be helpful to explain.*

Response: We thank the reviewer for their helpful feedback and we have made this clear in the caption of Fig. 1.

Response: - *Minor: line 388. Do the authors mean the Dynes formula?*

Response: Yes, we were referring to the “Dynes formula” and apologize for the typo (in the process of revising that paragraph, see response to reviewer 1, we removed the explicit reference to it).

Response: - *Minor: Eq. S12. I don't expect this to change any of the author's results qualitatively, but the form of the screened potential used is for double-gated devices, while the STM works are single-gated. The form of the potential would then be (obtained from solving Poisson's equation for vacuum - material at $z = 0$ - dielectric of thickness d and dielectric constant ϵ - metal gate), is $\frac{e^2}{2\epsilon_0 q} \frac{2}{1+\epsilon_r \coth(qd)}$ with $\epsilon_r = \epsilon/\epsilon_0$. This potential is less screened at a small q than the one the authors use.*

Response: We note the reviewer's comment on single vs. double gate screened devices. We do not expect that using the single-gated Coulomb potential will affect our results since we have carried out computations with both the un-renormalized TBG flat bands and the Coulomb interaction renormalized bands and find the same leading instability in either case for time-reversal symmetric

intervalley interactions (whether that interaction be from phonons or T-IVC fluctuations).

List of changes

1. We have corrected a factor of 2 difference between the coupling previously used in the linearized gap equation at T_c and the full gap equation at $T = 0$. Consequently, the eigenvalues in Fig. 3c are a factor of 2 smaller, the value of the coupling used for the T_c computation in Fig. 4 is a factor of 2 larger and the couplings used in the full gap equation are a factor of 2 smaller relative to the linearized gap equation V_c . Additionally, we have changed the value of the Lorentzian parameter in Fig. 2 as we found that the lowest V curve had a very small but positive condensation energy. By reducing the Lorentzian parameter by a factor of 4, we have now determined that all data at $T = 0$ corresponds to stable states with negative condensation energy. We also added the edge points of the Brillouin zone back into the Linearized gap equation computation used to generate Fig. 2C, which we had previously excluded for all linearized gap equation computations, and our T_c in this case consequently decreased. We noted this effect on our linearized gap computations in our supplementary materials.
2. We have explicitly referenced the Bogliobov Fermi surfaces in our abstract now and added an additional statement near line 67 emphasizing how a combination of the symmetries and band structure makes a relatively unconventional like our A_2 pairing a realistic candidate. In the conclusion, we now also explicitly reiterate that the energetic dominance of the A_1 superconductor is surprising given its interband nature.
3. We have changed the all phrases which reference “chiral d-wave state” to “chiral p-wave state”.
4. We have added an explicit reference to Table II already in the introduction to direct the interested reader directly to the table in the manuscript where the order parameters for the correlated insulators are listed.
5. We have added a comment at line 146 stating how the classification of spin triplet pairings changes in the presence of strain.
6. We have clarified in the caption of table 1 that $\hat{\chi}_k$ refers to a real and symmetric matrix.
7. We have corrected the missing sign in the header of table 1 as pointed out by reviewer #1.

8. We have added the labels to $\sigma^{\alpha\alpha}$ to line 150 as pointed out by reviewer #1.
9. We have removed the comment “(note that $\nu = -1$ is inconsistent with C_{3z})” in line 253 of the previous version of the manuscript.
10. We have clarified in the caption of figure 2 that the K , K' , and Γ points refer to the high symmetry points of the moiré Brillouin zone.
11. We have clarified in line 336 that $\nu = 2$ in the context of the band structures in Fig. 2 refers to the interaction renormalization and not the chemical potential (which is set to $\nu = 2.5$ as stated in Fig. 2’s caption).
12. We have changed the final “(b)” in the caption of Fig. 3 to “, see (a)”. We further added a label to the horizontal axis of Fig. 3(c).
13. We have rephrased the sentence in line 401 to emphasize our conclusion that our proposed pairing will produce a dI/dV curve that cannot be fit by a conventional fully gapped, s-wave pairing.
14. We have clarified on line 412 that ρ acts on the sublattice basis.
15. On line 474, we have commented on the implications for pairing mechanism if the normal state is ultimately discovered to be spin polarized or spin-valley locked, in response to reviewer #3’s comments.
16. We have extended the caption of Table II to explicitly state the relation to Fig. 4 already in the caption rather than just the main text.
17. On line 488, we have explicitly stated what the K-IVC, SLP+, and SLP- orders are.
18. On line 494, we have clarified that $\bar{\lambda}$ refers to the given order parameter in the microscopic sublattice basis.
19. On line 534, we have clarified that the precise condition where the E_1 pairing is nodal.
20. We added one sentence in the last paragraph of the discussion section, listing Andreev reflection as another promising future research direction.

21. We have removed our previous schematic band structure plot from our SI since a version of the same figure has been added to main text Fig. 1.
22. We have clarified in the caption of Fig. 1 what BI and BEC stand for.
23. We have added a Methods section containing the proof of Eq. (7) and technical details on how the phonon-induced electron-electron interaction arises (removed from the main text).
24. Finally (and unrelated to the reports), we have added an additional affiliation to the last author.

Reviewers' Comments:

Reviewer #1:

Remarks to the Author:

The authors have comprehensively addressed my comments and have made appropriate changes to the manuscript. I think that it is now easier to digest for the readership of Nature Commun. I still have one comment concerning Sec. II.F, including Table II and Fig. 4: While everything becomes clear if one carefully reads the main text all the way through, I still find the organization of information not optimal. For example, Table II contains four cases and Fig. 4 contains four panels, but these panels do not correspond to the rows of the table. Rather, three of them do, but in a different order, while N-IVC fluctuations do not appear in the table. The N-IVC order is only defined later in line 527.

There is also a typo in line 489.

I would like to briefly comment on the other two reports and the authors' responses. I do not agree with the second reviewer that the paper is too technical for the broad readership of Nature Communications. What matters, in my opinion, is whether the results are original, novel, and relevant and thus advance science. I think that they do and that the other reviewers agree. If the results require a certain level of technical explanation then it is necessary to include this explanation to allow others to check and extend the results. Moreover, if the advances themselves in part pertain to the theoretical description, like in this case, then the theoretical description needs to be presented.

My impression is that the authors have carefully answered the comments made by reviewers 2 and 3. In their last (minor) comment, the second reviewer pointed out an inconsistency in the chosen filling fraction, which is clear from the second paragraph of Sec. II.D. Also, in their last (minor) comment, the third reviewer noted that the authors used an inapplicable expression for the screened Coulomb interaction. The reviewers and the authors agree that these deviations do not affect the results qualitatively but in my view they still slightly mar the paper. Why not do it right if that is possible? If that would take several weeks of work it is probably not worth doing. This is a suggestion of minor importance, not a recommendation.

As noted in my first report and not repeated here, there are several remarkable results concerning the superconducting pairing states in twisted multilayer graphene, which is at the focus of current research. The nature of the observed superconducting states is not well understood, and the manuscript takes a large step in understanding them. I therefore find the manuscript suitable for publication in Nature Communications, after the authors have considered the above comments.

Carsten Timm

Reviewer #2:

Remarks to the Author:

I consider that the reply to my comments as well as the changes performed in the revised version of the manuscript address the main issues that I raised. Then, I recommend the publication of the manuscript in its current form.

Reviewer #3:

Remarks to the Author:

I thank the authors for their careful revisions and answers to all of my questions.

The authors have answered all of my questions, and I do not have any further comments.

I am happy for the paper to be published in its present form.

Resubmission of NCOMMS-23-16329:

**“Nodal band-off-diagonal superconductivity in twisted graphene
superlattices”**

by Maine Christos, Subir Sachdev, and Mathias S. Scheurer

(Dated: September 27, 2023)

Response to Reviewer # 1:

Reviewer: *The authors have comprehensively addressed my comments and have made appropriate changes to the manuscript. I think that it is now easier to digest for the readership of Nature Commun. I still have one comment concerning Sec. II.F, including Table II and Fig. 4: While everything becomes clear if one carefully reads the main text all the way through, I still find the organization of information not optimal. For example, Table II contains four cases and Fig. 4 contains four panels, but these panels do not correspond to the rows of the table. Rather, three of them do, but in a different order, while N-IVC fluctuations do not appear in the table. The N-IVC order is only defined later in line 527.*

Response: We agree with the referee that the presentation of Table II and Fig. 4 with respect to the N-IVC fluctuation mechanism is not optimal. We have therefore rearranged the presentation in Section II.F so that the strong coupling states in Table II are now discussed separately from the N-IVC state. Furthermore, we now introduce a description of the N-IVC state along with our discussion of the band projected strong coupling orders after referencing Table II and before referring to Fig. 4. This streamlines the presentation and a reader does not have to read all the way through to understand the relation between Table II and Fig. 4.

Reviewer: *There is also a typo in line 489.*

Response: We thank the reviewer for pointing out this typo and have corrected it.

I would like to briefly comment on the other two reports and the authors' responses. I do not agree with the second reviewer that the paper is too technical for the broad readership of Nature Communications. What matters, in my opinion, is whether the results are original, novel, and relevant and thus advance science. I think that they do and that the other reviewers agree. If the results require a certain level of technical explanation then it is necessary to include this explanation to allow others to check and extend the results. Moreover, if the advances themselves in part pertain to the theoretical description, like in this case, then the theoretical description needs to be presented. My impression is that the authors have carefully answered the comments made by reviewers 2 and 3. In their last (minor) comment, the second reviewer pointed out an inconsistency in the chosen filling fraction, which is clear from the second paragraph of Sec. II.D.

Also, in their last (minor) comment, the third reviewer noted that the authors used an inapplicable expression for the screened Coulomb interaction. The reviewers and the authors agree that these deviations do not affect the results qualitatively but in my view they still slightly mar the paper. Why not do it right if that is possible? If that would take several weeks of work it is probably not worth doing. This is a suggestion of minor importance, not a recommendation.

Response: We first thank the reviewer for stating that we “carefully answered the comments by reviewers 2 and 3” and that he agrees with us that the paper is not too technical for Nature Communications.

Concerning the last minor comment about the form of the Coulomb potential we use when computing the normal state band structure used in our fluctuation-mediated superconductivity numerics: given the time constraint of 2 weeks given for us to respond, we do not believe we would be able to remake our figures within the given time. At the same time, since the nature of the leading superconducting state primarily depends on the analytic condition given in Eq. (7), we do not think that changing the expression for a screened Coulomb potential would result in any qualitative change of our results; we would expect changing the form of the Coulomb interaction may change either the band splitting or total bandwidth a little bit; however, as long as the couplings we use for our pairing potential are large enough to overcome the band splitting, we have no expectation that the leading instabilities we find would change nor any of their key features.

Reviewer: *As noted in my first report and not repeated here, there are several remarkable results concerning the superconducting pairing states in twisted multilayer graphene, which is at the focus of current research. The nature of the observed superconducting states is not well understood, and the manuscript takes a large step in understanding them. I therefore find the manuscript suitable for publication in Nature Communications, after the authors have considered the above comments.*

Response: We thank the reviewer for their reading and response to our previous comments and for their positive recommendation.

Response to Reviewer # 2:

Reviewer: *I consider that the reply to my comments as well as the changes performed in the revised version of the manuscript address the main issues that I raised. Then, I recommend the publication of the manuscript in its current form.*

Response: We thank the reviewer for carefully reading our response to their previous comments and for their recommendation for publication.

Response to Reviewer # 3:

Reviewer: *I thank the authors for their careful revisions and answers to all of my questions. The authors have answered all of my questions, and I do not have any further comments. I am happy for the paper to be published in its present form.*

Response: We thank the reviewer for their reading of our revisions and response and for their acceptance of our current manuscript.

List of changes

1. We rearranged the discussion in Sec. II.F such that the N-IVC order is defined earlier in the text and one can clearly understand the relation between Table II and Fig. 4 without having to read the section all the way through.